# In-Depth Phenotyping of *PIGW*-Related Disease and Its Role in 17q12 Genomic Disorder

**DOI:** 10.3390/biom14121626

**Published:** 2024-12-18

**Authors:** Agnese Feresin, Mathilde Lefebvre, Emilie Sjøstrøm, Caterina Zanus, Elisa Paccagnella, Irene Bruno, Erica Valencic, Anna Morgan, Alberto Tommasini, Christel Thauvin, Allan Bayat, Giorgia Girotto, Luciana Musante

**Affiliations:** 1Department of Medicine, Surgery and Health Sciences, University of Trieste, 34127 Trieste, Italy; alberto.tommasini@burlo.trieste.it (A.T.); giorgia.girotto@burlo.trieste.it (G.G.); 2SoFFoet, Société Française de Fœtopathologie, 75015 Paris, France; mathilde.becmeur-lefebvre@chu-orleans.fr; 3UF de Foetopathologie, CHU d’Orléans, 45100 Orleans, France; 4Department of Epilepsy Genetics and Personalized Medicine, Danish Epilepsy Center, 4293 Dianalund, Denmark; emiliebangsjostrom@gmail.com (E.S.); abaya@filadelfia.dk (A.B.); 5Institute for Maternal and Child Health, IRCCS “Burlo Garofolo”, 34137 Trieste, Italy; caterina.zanus@burlo.trieste.it (C.Z.); elisa.paccagnella@burlo.trieste.it (E.P.); irene.bruno@burlo.trieste.it (I.B.); erica.valencic@burlo.trieste.it (E.V.); anna.morgan@burlo.trieste.it (A.M.); 6Inserm—UB UMR 1231 GAD “Génétique des Anomalies du Développement”, Fédération Hospitalo-Universitaire-TRANSLAD, 21000 Dijon, France; christel.thauvin@chu-dijon.fr; 7Department of Regional Health Research, University of Southern Denmark, 5230 Odense, Denmark; 8Department of Drug Design and Pharmacology, University of Copenhagen, 2100 Copenhagen, Denmark; 9Department of Paediatrics, Danish Epilepsy Center, 4293 Dianalund, Denmark

**Keywords:** *PIGW*, GPIBD11, heart malformation, epilepsy, fetus, 17q12 genomic disorder

## Abstract

Glycosylphosphatidylinositol (GPI) biosynthesis defect 11 (GPIBD11), part of the heterogeneous group of congenital disorders of glycosylation, is caused by biallelic pathogenic variants in *PIGW*. This rare disorder has previously been described in only 12 patients. We report four novel patients: two sib fetuses with congenital anomalies affecting several organs, including the heart; a living girl with tetralogy of Fallot, global developmental delay, behavioral abnormalities, and atypic electroencephalography (EEG) without epilepsy; a girl with early-onset, treatment-resistant seizures, developmental regression, and recurrent infections, that ultimately passed away prematurely due to pneumonia. We also illustrate evolving facial appearance and biochemical abnormalities. We identify two novel genotypes and the first frameshift variant, supporting a loss-of-function pathogenic mechanism. By merging our cohort with patients documented in the literature, we deeply analyzed the clinical and genetic features of 16 patients with *PIGW*-related disorder, revealing a severe multisystemic condition deserving complex management and with uncertain long-term prognosis. We consider the role of *PIGW* within the critical 17q12 region, which is already associated with genomic disorders caused by deletion or duplication and characterized by variable expressivity. Finally, we discuss *PIGW* dosage effects and a second hit hypothesis in human development and disease.

## 1. Introduction

Glycosylphosphatidylinositols (GPI) are membrane glycolipids that anchor various proteins—such as enzymes, adhesion molecules, receptors, and complement regulatory proteins—to the cell surface. These proteins are Glycosylphosphatidylinositol Anchored Proteins (GPI-APs) [1]. The GPI-APs biosynthetic pathway is a multi-step process involving at least 16 steps requiring several distinct proteins encoded by Phosphatidyl Inositol Glycan (*PIG*) or Post GPI Attachment to Proteins (*PGAP*) genes. In humans, the GPI-APs pathway critically contributes to several developmental processes, including embryogenesis, neurogenesis and immune response [2,3,4,5].

Pathogenic variants in *PIG/PGAP* genes cause a group of clinically and genetically heterogeneous conditions, collectively known as GPI biosynthesis defects (GPIBDs), associated with a broad clinical spectrum, including intellectual disability ranging from mild to profound, global developmental delay affecting speech and motor skills, epilepsy with often early-onset and treatment-resistant seizures, behavioral and psychiatric comorbidities, movement disorders, congenital anomalies, and dysmorphic features [6,7,8]. *PIGW* (OMIM #610275), located on 17q12, encodes 504 amino acids inositol acyltransferase, characterized by 13 transmembrane domains.

Biallelic *PIGW* variants lead to an autosomal recessive condition,“hyperphosphatasia with mental retardation syndrome 5” or GPIBD11 (OMIM #616025), which was recently recognized as an ultra-rare condition. Only 12 patients with homozygous or compound heterozygous variants, including single nucleotide variants and a gene deletion, have previously been described [4,9,10,11,12,13,14,15]. Reported patients include five fetuses with multiple congenital anomalies and seven live-born children with global developmental delay, epilepsy, hypotonia, dysmorphic features, congenital anomalies, and elevated phosphatase serum alkaline (ALP) [4,9,10,11,12,13,14,15]. However, the phenotype remains ill-defined.

We report four new patients with genetically confirmed *PIGW*-related disorders encompassing two previously unreported genotypes, one of which includes the first documented frameshift variant. We also present a comprehensive review of previously reported patients’ clinical, molecular, and biochemical features. By merging these independent cohorts, we have deep-phenotyped and expanded the antenatal, post-mortem, and postnatal clinical spectrum of the *PIGW*-related disorder.

Finally, showing the multi-systemic role of *PIGW* in human development and disease and the related pathogenic mechanism, we interrogate the role of the gene within the recurrent genomic disorders associated with the 17q12.2 locus.

## 2. Materials and Methods

### 2.1. Patients

P1 was recruited at the Institute for Maternal and Child Health “Burlo Garofolo” in Trieste, Italy. A multidisciplinary team consisting of a pediatrician, pediatric neurologist, cardiologist, biologist and geneticist specialized in rare diseases was involved in the evaluation, diagnostic process, and follow-up. The diagnostic procedure included the discussion of whole exome sequencing (WES) data in the context of phenotypic data at interdisciplinary meetings [16]. Three additional novel patients were identified through GeneMatcher and Société Française de Fœtopathologie (SoFFoet) [17,18]. Clinicians and geneticists collected clinical and genetic details with a standardized form. We conducted our investigations according to the principles of the Declaration of Helsinki and local Institutional Ethical Committees and obtained informed consent from the participating subjects. We obtained permission to publish the photographs of all subjects shown in Figure 1.

### 2.2. WES Analysis, Interpretation and Validation

Singleton/Trio WES were performed at the respective institutions. Genomic DNA was extracted from peripheral blood. The target enrichment kit and WES statistics are detailed in Appendix A. Data processing, including sequence alignment to GRCh37/hg19, variant filtering and prioritization, predicted functional impact, inheritance and validation, were performed as previously reported and summarized in the Appendix A. We confirmed and segregated analysis of variants using Sanger sequencing.

### 2.3. Immune Cytometry

CD16 is a GPI-anchored protein highly expressed on the surface of natural killers, neutrophils, monocytes, and macrophages. To analyze CD16 expression, 100 µL of heparinized blood samples were surface stained with PE-conjugated anti-CD16 antibodies (Miltenyi Biotec, Bergisch Gladbach, Germany) and incubated for 20 min at room temperature. The samples were investigated using the MACSQuant Analyzer 10 flow cytometer (Miltenyi Biotec, Bergisch Gladbach, Germany) and processed with FlowLogic software, https://flowlogic.software/ last (accessed on 4 December 2024).

### 2.4. Graphs

We created histograms and pies (Figure 3) with Excel (Microsoft) and the “Alluvial plot” (Figure 3e) online (https://www.bioinformatics.com.cn/plot_basic_alluvial_plot_017_en, accessed on 4 December 2024).

### 2.5. Search Method for Literature Review

We searched MEDLINE (PubMed) with the keywords “GPI”, “GPIAP” or “glycosylphosphatidylinositol-anchored protein” in combination with “PIGW”. We further reviewed every relevant reference in the acquired articles which we did not find in the MEDLINE search. We considered articles written in English and published between 2010 and July 2024. We gathered only cases with a confirmed molecular diagnosis.

## 3. Results

We identified a total of four novel patients with genetically confirmed *PIGW*-related disorders. These include two sib fetuses and two live-born girls.

### 3.1. Clinical Reports

The proband 1 (P1) was a 4-year-old girl, second-born of a healthy, non-consanguineous couple from northern Italy, who had one previously unexplained miscarriage and a healthy boy. The girl’s family history was unremarkable. After a spontaneous conception, an increased risk for trisomy 21 (1/94) resulted in a combined first-trimester screening test. We detected an overriding aorta at the second-trimester ultrasound (US) at 20 weeks of gestation (wgs). After genetic consultation, we performed amniocentesis with non-conclusive karyotype, FISH investigation for 22q11.2 deletion, and genomic microarray. The second-level US followed the pregnancy for cardiac defect without the detection of other anomalies. The baby was spontaneously delivered at 37 + 6 wgs. Birth weight was 2700 g (90 percentile, pct). APGAR was 9/9. Newborn screenings (hearing and metabolic) were normal. At birth, an echocardiogram confirmed a complex cardiac malformation type tetralogy of Fallot, with ventricular septal defect, without obstruction but with altered pressures in the pulmonary artery, and with an atrial septal defect, ostium secundum type. The patient underwent medical treatment with furosemide until the surgical correction at two months of life. Subsequently, she presented global developmental delay. Genetic evaluation at 21 months of life disclosed peculiar facial features: squared face with a broad forehead, depressed nasal root with a short nose, thin upper lip, occasional mild protruding tongue, and diastema. Growth parameters (height: 83 cm (52 pct), weight 10 kg (15 pct), head circumference 47.7 cm (70 pct)) were average. Neuropsychiatric and neuropsychologic evaluations confirmed a global developmental delay with gross motor difficulties and verbal language delay. Clinical re-evaluations at 38 (Figure 1(a1)) and 45 months of age (Figure 1(a2)) described more evident facial dysmorphisms, including a coarse face with a prominent forehead, depressed nasal root, marked philtrum, and widened mouth with thin upper lip. In the profile view, we also observed anteverted, slightly low-set ears, a short nose, a protruding philtrum, and micrognathia (Figure 1(a3)). There were no hand anomalies, but the left toe was low-set, and the subject presented overcrowding of the toe (Figure 1(e2)). We also noted a unilateral inguinal hernia. Neurologic re-evaluation confirmed a global developmental delay, with absent speech, limited comprehension, and immature gross and fine motor skills. The objective neurologic examination revealed that the patient had low muscle mass but normal strength. No clinically recognizable epileptic episodes were reported. We noted behavior abnormalities with poor gaze engagement, limited pointing, and deficit attention. EEG during sleep proved: (a) the absence of typical elements of physiological organization that differentiate phases of drowsiness/sleepiness from phases of deeper sleep; (b) the presence of diffuse rapid activity that increases in amplitude and becomes dominant as sleep onset progresses; and (c) the presence of multifocal paroxysmal abnormalities (Figure 1(e4)). Given the absence of clinically recognizable epilepsy and limited patient collaboration, the EEG was recorded during sleep only at follow-up. Surgical treatment for correction of the right inguinal hernia was performed before the age of four. Odontostomatological evaluation revealed tooth agenesis and maxillofacial CT confirmed the inclusion of elements 82–83 and the absence of elements 35 and 45. Biochemical investigations pointed out hypophosphatasia with increased plasma values of ALP (519 U/L) at 45 months of age. Flow cytometry documented a near-absent expression of CD16 on the patient’s neutrophils (Figure 1(e3)). Comprehensively, immunophenotyping showed a GPI-anchored protein deficiency. Brain MRI, abdominal US, ophthalmological, ear, nose and throat and audiological evaluations were unremarkable. Personalized follow-up is ongoing at the referral pediatric center for rare diseases.

Proband 2 (P2) was the second-born child of healthy Danish parents, with a healthy older brother and later a healthy younger brother. Her mother had well-managed hypothyroidism throughout the otherwise uneventful pregnancy. The patient was born at 37 + 5 wgs. Birth weight was 4040 g. At birth, she was able to breastfeed. Due to persistent jaundice and lethargy, she was treated with phototherapy at the maternity ward. At four days old, she was examined by a pediatrician due to folds on both ears. Having dismissed the suspicion of a syndrome, the family was advised to consult an ear, nose and throat- specialist. The facial appearance developed gradually after birth (Figure 1(b1–b3)). At five months, the suspicion of a syndrome was raised again due to delayed motor development and distinctive facial appearance: she was slightly dysmorphic, with dolichocephaly, hypotelorism, large ears, a flat nasal bridge, and thin feathery hair. She also exceeded height and weight curves by more than three standard deviations. At six months, an abdominal US revealed bilateral hydronephrosis, which was treated with prophylactic antibiotics. Concomitantly, she was hospitalized several times a month due to frequent respiratory infections. By 10 months, she could roll back to front and grasp objects with her hands, but she still had limited head control. However, epileptic spasms began at 10 months, leading to developmental regression. The initial seizures were characterized by a distant gaze and cyanosis around the lips, accompanied by a series of flexions in the trunk and upper extremities, lasting for 2–3 s each, occurring consecutively, and associated with vomiting. The first EEG was described as hypsarrhythmic. She lost head control, the ability to roll, and the use of her hands. By age 3, she exhibited profound global developmental delay, with minimal abilities beyond briefly grasping objects. She had no eye contact, no speech, and experienced difficulties swallowing. Her seizures were challenging to manage; we tried several anti-epileptic drugs, prompting the initiation of a ketogenic diet.

Additionally, she had elevated alkaline phosphatase levels: ALP was measured 12 times from 9 to 42 months of age, and it was always elevated between 610–940 U/L (reference 134–518 U/L). She also required glasses due to hypermetropia. Approximately one month prior to her death, we documented two additional seizure types: (1) vertical tonic eye deviations persisting for several minutes and (2) focal tonic seizures with impaired awareness lasting more than three minutes, characterized and accompanied by eye deviation and apnea. The patient passed away at 3 years and 10 months of age due to asystole-related to pneumonia.

Proband (P3) was a 19 + 5 wgs fetus born of a healthy consanguineous couple from France, who had a previously unexplained miscarriage. After a spontaneous conception, an antenatal ultrasound identified short, long bones with femoral bowing, short hands and feet, cardiopathy and Blake’s pouch cyst. Because of the severe prognosis, termination of pregnancy (TOP) was established at 19 + 5 wgs. A fetal autopsy identified a female fetus with nuchal oedema, microcephaly, micromelia with brachyphalangy (Figure 1(c1)) and facial dysmorphism, including hypertelorism, short nose and low set ears (Figure 1(c2,c3)). Postmortem X-rays showed platyspondyly, shortening of long bones, short ribs and enlarged metaphysis (Figure 1(e1)). Internal examination disclosed bilateral renal dysplasia, overriding aorta and small brain with enlarged ventricles and cerebellum hypoplasia.

A few months later, the couple had a third pregnancy (P4). A first-trimester ultrasound identified omphalocele, shortening of the long bones and cleft palate. The parents asked for a TOP at 14 + 3 wgs. A fetal autopsy identified a male fetus with micromelia, duplication of right hallux, omphalocele and anal atresia (Figure 1(d1)). Facial dysmorphism included turricephaly, a short nose, and a long and smooth philtrum (Figure 1(d2,d3)). Postmortem X-rays revealed shortening of the long bones, platyspondyly, short ribs and brachyphalangy. Internal examination disclosed significant great artery asymmetry and bilateral renal dysplasia.

### 3.2. Molecular Findings

WES in P1 identified two *PIGW* missense variants, found in a heterozygous state in the healthy father (c.106A>G; p. (Arg36Gly)) and mother (c.1227T>G; p. (Cys409Trp)). The variants were classified as likely pathogenic (see Appendix A). We did not detect any additional (likely) pathogenic variants that could explain the proband’s entire or partial phenotype. In particular, homozygous p.(Arg36Gly) has been previously reported in three fetuses from two unrelated families [13,15] and described as pathogenic in the HGMD database, while p.(Cys409Trp) has never been reported.

We evaluated the functional impact of the substitutions using several in silico tools, which predicted these variants as deleterious with a CADD Phred score of 23.4 and 26.7, respectively. The substitution in position 36 of PIGW affected an amino acid located in the protein’s first transmembrane domain. In contrast, the second change was in a protein region between the 10th and 11th domains (Figure 2).

WES in P2 identified compound heterozygous *PIGW* variants (c.50C>T, p.(Thr17Ile); c.1202del, p.(Leu401GlnfsTer6)) existing in heterozygous state in the healthy parents. The missense change was reported once in ClinVar (Variant_ID 1928004), and the frameshift was novel and never before reported. Notably, the missense variant was absent from gnomAD, while the frameshift variant was reported only in the heterozygous state with a very low allele frequency (Appendix A). The variants were classified as likely pathogenic and pathogenic, respectively (see Appendix A). The substitution in position 17 of PIGW affected an amino acid in the N-terminal facing the endoplasmic reticulum.

In P3, singleton WES analysis identified a known likely pathogenic homozygous missense variant in *PIGW* (c.106A>G; p.(Arg36Gly)). Sanger sequencing confirmed the presence of the p.(Arg36Gly) in a homozygous state in the affected sibling, P4. The healthy parents were confirmed to be carriers of the variant.

All patients had preliminary, negative micro-array analysis.

**Figure 2 biomolecules-14-01626-f002:**
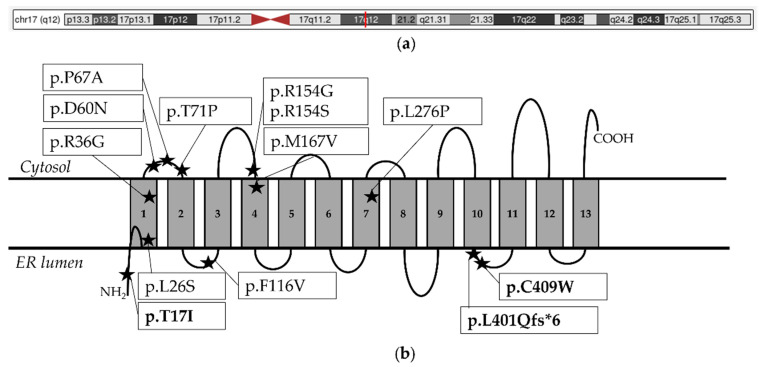
*PIGW* reported variants (**a**) Chromosome 17 representation, indicating the location of *PIGW* gene in red, from UCSC Genome Browser on Human (https://genome.ucsc.edu/index.html, accessed on 28 August 2024), The red line in the figure caption corresponds to the genomic region 17q12.2, which is critical for recurrent deletion/duplication and involves various contiguous genes, including *PIGW*, Numbers 1–13 refer to PIGW transmembrane domains; (**b**) schematic diagram of the PIGW protein (NP_848612.2), showing the position of the reported variants indicated with stars (shortened amino acids nomenclature). Novel variants in our patients are marked in bold.

### 3.3. Literature Review

Eight selected papers reporting 12 *PIGW* patients belonging to 10 unrelated families have been evaluated [4,9,10,11,12,13,14,15] and included in the study. Family histories, molecular investigations, results and interpretations, and detailed description of clinical and biochemical findings have been collected and summarized in Table 1, together with the patients reported here.

The molecular, clinical and biochemical findings related to the totality of *PIGW* patients from the literature and the current study have been analyzed together.

Twelve missense single nucleotide variants, one frameshift variant, and one gene deletion are reported. Most of the variants are in the transmembrane domains of the protein (9/12, 75%). All single nucleotide variants are represented in the schematic protein according to their position (Figure 2).

We extrapolated information concerning the setting (pre-/post-natal) of the first medical referral and molecular diagnosis determination (post-mortem/natal). The first referral was at the early stages of gestation during the first trimester in most cases (7/9, 77.7%). 9/16 (56.2%) had at least one prenatal finding (Figure 3a), and seven out of the nine (77.7%) had multiple congenital anomalies (Figure 3b). All fetuses with multiple congenital anomalies were diagnosed following a fetal pathological examination, which permitted an extension of the phenotype description (Figure 3c). No prenatal diagnosis was performed in any case (Figure 3d).

After reviewing all the clinical descriptions of patients, we grouped most of the signs and symptoms into major categories: congenital anomalies, facial dysmorphisms, neurodevelopmental disorder/neurological findings including epilepsy, EEG abnormalities and behavior, and biochemical findings. Associations between genotypes and main categories have been taken into account (Figure 3e).

**Figure 3 biomolecules-14-01626-f003:**
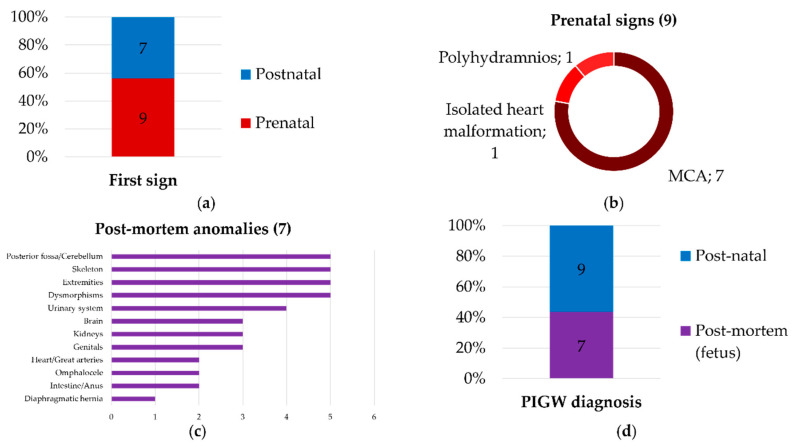
Referral, diagnostic setting and presentation (**a**) Histogram showing the number of patients based on the setting (prenatal/postnatal) of the first sign at the time of medical referral; (**b**) pie chart representing the distribution of presentations in patients with prenatal signs, numbers refer to the patients; MCA: multiple congenital anomalies; (**c**) histogram showing the number of fetuses having anomalies in the listed systems/organs; (**d**) histogram representing the number of patients according to the setting (post-mortem/postnatal) at the moment of molecular diagnosis with *PIGW* variant identification; (**e**) alluvial plot showing, from left to right: column 1: setting of onset/referral (PostN: postnatal, PreN: prenatal), column 2: *PIGW* allele 1, column 3: *PIGW* allele 2, (shortened amino acids nomenclature for *PIGW* variants, number in brackets refer to the patients; color correspondence between column 2 and 3 refers to patients), column 4: presentation according to major phenotype categories (major phenotype categories: biochemical findings (green), congenital anomalies (red), EEG abnormalities without epilepsy (light blue); facial dysmorphism (yellow), neurodevelopmental/neurological and EEG abnormalities with epilepsy (dark blue), (**f**) histograms showing the number of patients with congenital anomalies (in red, on the left), neurodevelopmental/neurological/behavioral features (dark blue, in the middle and top), EEG abnormalities (light blue, in the middle and bottom), facial dysmorphism (in yellow, on the right and top), biochemical findings (in green, on the right and in the middle), other clinical features (in orange, on the right and bottom). Novel features presented in our patients are marked with a triangle with red outline. Legend: T1: first trimester; PreN: prenatal; PostN: postnatal; suba sp: sub-arachnoid spaces; wm: white matter; ToF: Tetralogy of Fallot; sp: spams; Gen.: generalized; sy: syndrome.

## 4. Discussion

By merging data from novel and already reported patients, we analyzed a cohort of 16 *PIGW* patients, focusing on molecular, clinical and biochemical findings.

### 4.1. New Clinical Findings

New findings from the herein-described patients emerged. We reported for the first time three patients with cardiac and great artery malformations. P1 exhibited an antenatally detected overriding of the aorta within a tetralogy of Fallot, requiring neonatal surgery. P3 and P4 exhibited an overriding aorta and great artery asymmetry, respectively. Before this occurrence, only a patent ductus arteriosus was recorded in a child [10]. After excluding alternative causes, we questioned the role of *PIGW* variants in cardiac organogenesis. First, PIGW was expressed in the heart muscle (https://www.proteinatlas.org/ENSG00000277161-PIGW/tissue#rna_expression, accessed on 28 August 2024). In addition, cardiac anomalies were observed in patients with altered GPI-biosynthesis pathways. For example, biallelic variants in *PIGL* portrayed a syndromic association, including congenital heart disease, with coloboma, ichthyosiform dermatosis, mental retardation and ear anomalies (CHIME) [19,20]. In some patients, pathogenic variants in *PIGA* were associated with cardiac anomalies, especially atrial septal defects [21,22]. Although cardiomyopathy was previously reported as an overlooked feature and potentially the cause of early demise in *PIG/PGAP* genes, none of the *PIGW* patients in this paper were diagnosed with cardiomyopathy [23,24].

Odontostomatological evaluation in P1 indicated tooth agenesis. Considering the sporadic presentation in the family, the absence of alternative etiology and the already described widely spaced teeth [10,14], tooth agenesis could be a new finding.

An additional unique skeletal finding was a duplicated hallux found in P4. Skeletal and extremity anomalies were frequent in the *PIGW* cohort. Some fetuses presented shortening of the long bones, femoral bowing, short ribs, enlarged metaphysis and platyspondyly. Minor anomalies such as brachypahalangy/clinodactyly, thumb and hallux anomalies were also common. P2 showed early post-natal overgrowth. Although additional evidence is needed, overgrowth and tooth agenesis have been reported in at least one patient with a pathogenic *PIGA* variant, supporting a potential role of GPI-related pathways in skeletal and dental development [21]

P4 exceptionally presented anal atresia. Rarely, intestinal anomalies have been described, with only one fetus showing intestinal malrotation [15].

Otherwise, congenital anomalies in our patients confirmed the already reported frequent kidney dysplasia, genitourinary anomalies, brain and posterior fossa malformations and wall defects [4,13,15].

We also provide the first documented evolution of phenotype in P1 and P2, with slight similarities concerning the facial appearance, especially concerning the nose (depressed nasal root), the philtrum (marked philtrum) and the mouth (widened, down slanting, thin upper lip).

In addition, P1 was the first patient with recorded EEG abnormalities without clinically evident seizures. However, she presented with global developmental delay, which is a key finding in all *PIGW* children, and with behavioral anomalies, which have been less consistently reported.

Finally, we described P2, a girl with severe neurological presentation, developmental regression, and early-onset, progressive, uncontrollable epilepsy. Postnatal neurological presentation with varying severity was common in the previously reported cohort, and regression posed a differential diagnosis with Rett/likely Rett syndrome [14]. However, we highlight here the severity of P2’s natural history, who passed away at the age of 3 due to asystolia in the context of pneumonia. Recurrent respiratory infections have been documented in other patients, including one who developed epileptic clusters in West syndrome during the course of pneumonia at 23 months of age, with subsequent worsening of epileptic episodes during fever and infections [10,11,14]. In addition, one patient had frequent aspiration pneumonia [4].

Although a *PIGW* diagnosis was not ascertained, the index patient’s brother, reported by Fu and colleagues as having a developmental delay, died at the age of 7 months because of recurrent pulmonary infection [11]. P2 is the first *PIGW* patient to die within the first years of life, and we highlight the risk of death in cases of severe respiratory infection. Previously, some authors have stated that patients with GPI-ADs, especially *PIGA*-encephalopathy, were at risk of premature death due to the development of cardiomyopathy [23].

We noticed that in two families (P1, P3–P4), an early unexplained miscarriage was reported. Although present data are insufficient to advance the hypothesis of human embryonic lethality in case of biallelic *PIGW* variants, it might represent the extreme phenotype of the spectrum.

The genotype-reversal phenotype documented high serum ALP in both P1 and P2, supporting the correlation between biallelic *PIGW* variants and this candidate biomarker. In addition, P1 showed decreased GPI-APs, advancing the clinical interpretation of the novel genotype, and expanding the biochemical phenotype.

### 4.2. A Continuous and Broad Phenotypic Spectrum

Although only recently described, the ultra-rare *PIGW*-related disorder was associated with a heterogeneous syndromic presentation. On the one hand, as for P3 and P4, multiple structural anomalies affecting the brain and posterior fossa, genitourinary and skeletal systems, and diaphragm and abdominal wall characterized severe fetal presentations leading to TOP [13,15]. A partial phenotype description was available for them with unknown functional and neurologic outcomes. On the other hand, for P2, the first reason for postnatal medical referral was an early, severe neurodevelopmental condition with global developmental delay and early-onset epileptic encephalopathy or seizures [4,10,11,12,14]. P1 belonged to a likely intermediate clinical entity and supported the existence of an overlapping and continuous spectrum between early-onset presentation with multiple malformations and neurodevelopmental disorders. This study’s comprehensive data analysis highlighted a continuous and broad spectrum instead of a pleiotropic effect of *PIGW*.

Most patients had prenatal signs (9/16, 56.2%, Figure 3b), and several showed evident signs in the first trimester (7/9, 77.8%, Figure 3c), suggesting that *PIGW*-related disorders could appear very early due to anomalous fetal development. In most of these cases, TOP was required due to multiple congenital anomalies (7/9, 78%), which were correlated to a poor prognostic outcome even before the determination of an etiologic diagnosis. Half of the couples with an affected fetus (2/4, 50%) had a personal history of malformed fetuses, attesting to the recurrence of multiple congenital anomalies before the definitive diagnosis of recessive disease. All the cases with TOP underwent a fetal-pathological examination, allowing the confirmation of major anomalies and the description of minor anomalies and facial dysmorphisms. Previously, authors suggested a differential diagnosis with Fryns or Fryns-like syndrome [15]. The association of malformations resulted in multi-systemic involvement in all cases, affecting the brain (5/9) and heart (3/9). Including the newly described cardiac malformations, we demonstrate that *PIGW* disruption may affect the development of almost all major human systems, especially the brain, cerebellum, kidneys and genitourinary system, gastrointestinal system, skeleton, diaphragm and abdominal wall. In addition, we highlight the possibility of an isolated prenatal heart malformation.

To the best of our knowledge, there is no prenatal diagnosis of *PIGW*. This could mainly be related to the severity of fetal presentation, requiring a medical termination despite the precise etiology or, as in the case of our patient with an apparently isolated cardiac malformation, due to limited, first-level molecular investigation.

Two of the children, both with a diagnosis in the post-natal setting due to developmental concerns, had apparently isolated prenatal signs, including tetralogy of Fallot in P1 and polyhydramnios in the context of maternal diabetes [14]. Pediatric cases reached a conclusive diagnosis by WES or epilepsy-gene panels.

The comprehensive data collection in this review elucidated that most of the patients presenting with neurodevelopmental disorder/epilepsy had facial dysmorphisms and at least a concurrent congenital anomaly. Congenital anomalies could be major, especially involving the brain, brainstem or cerebellum, or minor, as in the case of hydronephrosis, skeletal, extremities anomalies and inguinal hernias. One of the patients described by Hogrebe et al. presented only with neurologic/neurodevelopmental issues with epilepsy and a large tongue without malformations [12]. However, her second cousin, harboring the same *PIGW* genotype, presented ankyloglossia and extremity anomalies, confirming that every *PIGW* genotype was associated with at least one congenital anomaly (Table 1, Figure 3e) [12]. Facial dysmorphisms are also very common.

To date, the *PIGW* sample size has been very limited. Even expanding the cohort up to sixteen characterized patients, the phenotypic spectrum may be biased, and it will likely be expanded in the future as additional cases are identified.

### 4.3. New Molecular Findings

We report on two novel genotypes, including two new missense variants and the first frameshift variant.

Only the p.(Arg36Gly) variant was recurrent in several patients, including P1, P3 and P4, in a homozygous (P3–P4) or composite heterozygous state (P1) and has been associated with all the phenotypic categories (Figure 3e).

This variant consisted of the substitution of the positive charge arginine in position 36 with a non-polar amino acid (glycine) within the first transmembrane domain of the protein (Figure 2a). Although the functional relevance of the p.(Arg36Gly) was not investigated, the variant has already been documented at homozygous state in three fetuses from two unrelated families [13,15]. In two sib fetuses with multiple anomalies, the variant was classified as likely pathogenic since the mutant amino acid was predicted to disturb the transport activity of the protein [13]. In the third fetus with early and severe clinical presentation, it was firstly classified as a variant of unknown significance (VUS) and finally considered to have a likely causative role [15].

The second missense variant c.1227T>G, p.(Cys409Trp), found in P1 only, consisted of the change of a nucleotide (thymine replaced with guanine), which involved the substitution of the conserved, polar, uncharged amino acid (cysteine) with an aromatic amino acid (tryptophan), and located between the tenth and eleventh transmembrane protein domain. This variant was novel and was not reported in *PIGW* patients nor disease/clinical databases (ClinVar, HGMD). Considering the clinically heterogeneous condition’s rarity, the novel variant in P1 was first classified as VUS. Biochemical insights attesting to decreased expression of GPIs provided further evidence for a likely causative role for the P1 genotype since pathogenic variants in genes coding for proteins involved in intermediate and late steps of GPI biosynthesis were primarily associated with increased plasma ALP levels.

The third missense variant, c.50C>T, p.(Thr17Ile), found only in P2, consisted of the substitution of a nucleotide (cytosine replaced with thymine), which involved the substitution of the polar amino acid (threonine) with the non-polar amino acid (isoleucine), located in the luminal topological domain. This variant was novel and had neither been reported in *PIGW* patients nor disease/clinical databases (ClinVar https://www.ncbi.nlm.nih.gov/clinvar/ accessed on 28 August 2024, HGMD (Qiagen) https://digitalinsights.qiagen.com/about-corporate/ accessed on 28 August 2024).

The variant c.1202del, p.(Leu401GlnfsTer6), found only in P2, was the first non-missense reported in *PIGW*. It consisted of a single nucleotide deletion leading to a shift of the reading frame and predicted to result in a prematurely truncating protein. The paucity of loss-of-function *PIGW* variants in the compound heterozygous state (and the absence in homozygosity) with a complete GPI deficiency could suggest that they were incompatible with life, advancing the hypothesis of a more extreme, early lethal phenotype, missing clinical attention and diagnosis. Unfortunately, knockout *PIGW* animals or cellular models to corroborate these hypotheses are lacking at the time of writing. Fang and colleagues described the first complete loss-of-function effect due to a heterozygous deletion of the entire *PIGW* gene in trans with a missense mutation [10]. The authors documented an inherited deletion, spanning about 1.4 Mb, involving likely additional contiguous genes in the 17q12 region (genomic coordinates not provided) [10]. The novel pathogenic frameshift variant in P2 thus supported a loss-of-function disease mechanism [10].

### 4.4. Considerations on PIGW Variants

This study identifies complex genotype-phenotype correlation with 12 missense variants, one frameshift variant, and one gene deletion combined in 10 private genotypes for 16 affected individuals. The families with *PIGW* patients were from different ancestries, including Asia, Europe, Northern Africa, and South America. Most patients had a *PIGW* variant in a homozygous state (11 from seven families). Consanguinity was reported in four families. Five unrelated individuals had compound heterozygous changes, including one presenting 17q12 deletion (breakpoints not reported) encompassing the *PIGW* gene [10]. All heterozygous parents and carrier family members were healthy.

The human PIGW-related GPI biosynthesis pathway is crucial in several developmental processes and embryogenesis [5]. GPI-APs play crucial roles in embryogenesis, and animal model studies have demonstrated that embryonic lethality occurred when there was complete GPI deficiency, as reported for the *Piga* gene knockout mice, which belongs to the same pathway as *PIGW* [25]. We can, therefore, speculate that cell surfaces lacking GPI anchoring proteins due to deficient *PIG/PGAP* gene activity are fundamental to correct human development.

As for other GPI genes, affected patients presented *PIGW* variants with likely partial or abolished protein activity [26]. Patient cells showed variably decreased levels of GPI-anchored proteins compared to controls, consistent with a hypomorphic or loss-of-function effect [12]. In line with these findings, in vitro functional studies elucidated that *PIGW* mutations reduced protein activity. In particular, Hogrebe and colleagues showed that the p.(Arg154Gly) had a reduced ability to restore GPI-APs in PIGW-deficient cells compared to wild-type protein [12].

Genotype-phenotype correlation is challenging in *PIGW*-related disorders. Firstly, the same genotype was associated with different manifestations. The totality of homozygous c.106A>G, p.(Arg36Gly) patients were fetuses (5) from three unrelated families presenting with multiple congenital anomalies involving great arteries/heart, brain, kidney and genitourinary system, skeleton, abdominal wall defects and dysmorphism. It was noteworthy that the final presentation also differed in the case of sibling fetuses, suggesting inter- and intra-familial clinical variability.

### 4.5. Challenging Genotype-Driven Follow-Up

In the post-genomic era of personalized medicine, getting a molecular diagnosis orients personalized access to care and follow-up. However, in the case of ultra-rare diseases with few described patients, such as *PIGW*-related disorders, a lack of available long-term follow-up or general guidelines makes patient management challenging. One of the main issues in *PIGW* children concerns the control of seizures. The great majority of pediatric patients experienced early-onset epilepsy (8/9, 88.9%), with several cases of partial control or uncontrolled epilepsy (5/8, 62.5%). P2 underwent different therapeutic approaches, including the initiation of a ketogenic diet, but they were all unsuccessful. P1 required a neonatal cardio-surgical intervention for tetralogy of Fallot, which was the first example of major surgery in the cohort of GPB11, and it took place before the determination of the diagnosis. P1 also underwent a minor surgical treatment for an inguinal hernia. The literature also described inguinal with or without umbilical hernia in other patients [9,11]. In P1, the benefits of an early diagnosis of a likely syndromic condition were evident in follow-up management. Considering the high prevalence of seizures, family educative counselling could help to detect and recognize at-risk episodes. Since seizures sometimes worsen during infection and fever, surveillance and prompt medical treatment for fever and respiratory infections could prevent potential severe medical complications. Odontostomatologic care for P1 was also planned, and multidisciplinary follow-up is ongoing at the referral center for rare diseases. No evidence-based prognostic information can be provided.

### 4.6. Perspectives

*PIGW* is located at the locus 17q12 within a critical genomic region. Recurrent, reciprocal contiguous gene syndromes are due to deletion or duplication around 1 or 2 Mb mediated by Non-Allelic Homologous Recombination (Appendix A).

The 17q12 deletion syndrome (OMIM #614527) is typically characterized by maturity-onset diabetes of the young type 5, renal cysts and Müllerian anomalies mainly associated with *HNF1B* haploinsufficiency (OMIM #189907) [27]. Besides the well-recognized role of *HNF1B*, among other genes encompassed in the genomic region, *LHX1* has been proposed as a possible modifier for earlier onset renal disease [28]. Considering the frequency of genitourinary involvement with kidney dysplasia and hydronephrosis, and the occurrence of Mullerian anomalies in patients with biallelic *PIGW* variants, *PIGW* could also be involved in these manifestations in larger microdeletions. Additional inconstant clinical features include dysmorphisms, developmental delay, intellectual disability, seizures and skeletal anomalies [29,30]. *LHX1* has been proposed as a candidate for neurodevelopmental findings and autism spectrum disorder, as well as in the microduplication syndrome [31]. The reciprocal duplication syndrome (OMIM #614526) is typically characterized by incomplete penetrance and more variable expressivity, including intellectual disabilities, behavioral abnormalities, and seizures [32]. To the best of our knowledge, variable expression in 17q12 genomic disorders is recognized without straightforward elucidation of causative genes and reasons underlying phenotypic differences. The prenatal presentation and the cohort of fetuses with heterozygous 17q12 deletion or duplication typically present hydronephrosis and kidney anomalies, while heart malformation and restricted growth have been described in an exiguous subgroup [33,34,35].

However, strict genotype-phenotype correlation is still missing, especially for heart malformation, seizure, dysmorphisms and skeletal anomalies.

Fang et al. reported the only compound heterozygous patients with a 1.4 Mb microdeletion encompassing *PIGW* and a contralateral *PIGW* missense variant, proposing biallelic *PIGW* alteration as the cause for neurologic presentation and epilepsy [10]. P2 was the first reported patient with a *PIGW* frameshift variant in trans with a missense one, with severe neurologic presentation, epilepsy and bilateral hydronephrosis. In both patients, a null *PIGW* allele together with a missense variant were associated to GPBD11. Experimental studies on some pathogenic *PIGW* missense variants suggested a reduced protein function, corroborating a likely loss-of-function *PIGW* disease mechanism. In a sub-group of individuals with 17q12 deletion/duplications, a second hit hemizygous *PIGW* alteration could contribute as a penetrance modifier for kidney anomalies, heart malformation, seizure, dysmorphisms and skeletal anomalies. Variants in other single genes, genomic imbalances, and alternative molecular mechanisms could probably explain clinical variability.

Further investigations are needed to clarify the role of *PIGW*, even in light of a more personalized prognosis determination in the case of early detection of 17q12 imbalances by first-tier prenatal molecular investigations in the setting of genomic microarray analysis and genomic screening.

## 5. Conclusions

The *PIGW*-related disorder is an ultra-rare, broad, continuous spectrum disorder characterized by congenital anomalies involving the heart and great arteries, severe neurodevelopment concern, and facial dysmorphisms. In this study, we highlighted the occurrence of EEG abnormalities without early-onset seizures, as well as early-onset, progressive and uncontrolled epilepsy and concomitant to frequent respiratory infections as a risk of severe, life-threatening medical comorbidity. Evoking a *PIGW* diagnosis based on clinical presentation has seemed highly challenging so far, enhancing the value of Next Generation Sequencing for precise diagnosis. In the case of a heterozygous pathogenic *PIGW* variant, microarray should be considered for 17q12 deletion. In contrast, the utility of intentional screening searching for pathogenic *PIGW* variants in individuals with 17q12 genomic disorder is still being determined. In the case of prenatal diagnosis of *PIGW*-associated disease, counselling will be based on the severity of congenital anomalies and the high risk of neurodevelopmental concerns and epilepsy. Biochemical markers, serum ALP and flow cytometry could supplement the genetic testing and contribute to *PIGW* variant classification in the post-natal setting. After diagnosis, dedicated management should include neurological, neurophysiological and developmental assessments; surveillance for epilepsy, respiratory infections and fever; growth monitoring, brain MRI, heart and abdominal US; ophthalmologic and odontostomatologic evaluations. Long-term follow-up and networking are fundamental for monitoring age-related comorbidities, defining prognosis, and orienting seizure treatments.

## Figures and Tables

**Figure 1 biomolecules-14-01626-f001:**
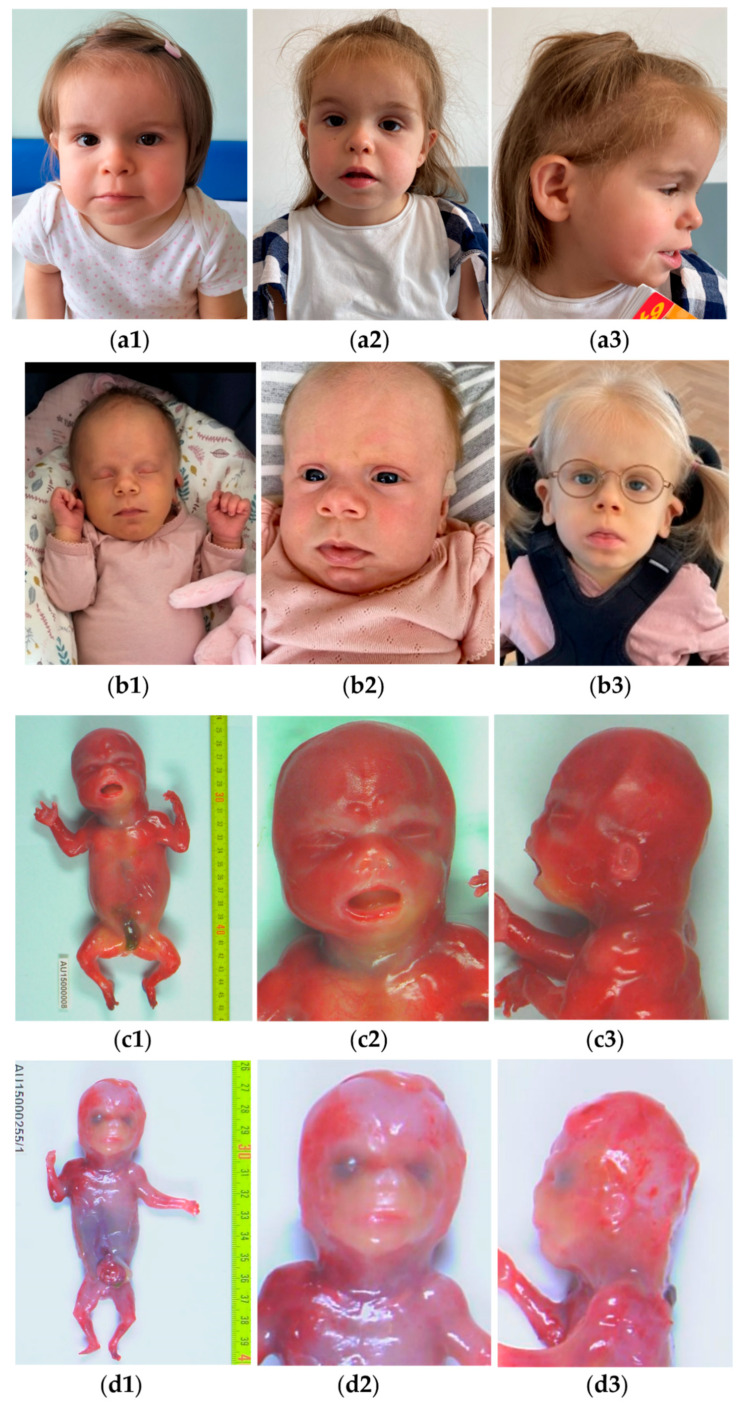
Phenotypic findings of novel *PIGW* patients. (**a1**) P1: in front view at 21 mth showing squared face with large forehead, depressed nasal root, short nose, and thin upper lip; (**a2**) P1: in front view at 38 mth showing coarse face with prominent forehead, depressed nasal root, marked philtrum, widened mouth with thin upper lip; (**a3**) P1: profile view evidencing anteverted, slightly low-set ears, short nose, protruding philtrum, micrognathia; (**b1**) P2: in front view at 20 dy showing long, large forehead in dolichocephaly, hypotelorism, bulbous nasal tip, and marked philtrum; (**b2**) P2: in front view at 2 mth showing prominent forehead, depressed nasal root, marked, protruding philtrum, thin upper lip; thin feathery hair; (**b3**) P2: in front view at 3.5 yrs showing prominent forehead, marked, protruding philtrum, widened mouth with thin upper lip, low-set, cupped ears with bilateral anteverted and down slanting helix and large lobes; (**c1**) P3: total body anterior view at 19 + 5 wgs showing micromelia; (**c2**) P3: in front view at 19 + 5 wgs showing hypertelorism and short nose; (**c3**) P3: profile view at 19 + 5 wgs showing low set ears; (**d1**) P4: total body anterior view at 14 + 3 wgs showing micromelia and omphalocele; (**d2**) P4: in front view at 14 + 3 wgs showing short nose, long and smooth philtrum; (**d3**) P4: profile view at 14 + 3 wgs showing turricephaly; (**e1**) P3: X-rays at 19 + 5 wgs showing short ribs and enlarged metaphysis; (**e2**) P1: feet showing low-set left toe; overcrowding of toe; (**e3**) P1: flow cytometry showing CD16 expression on patient’s neutrophils (red histogram) and donor’s neutrophils (blue histogram). (**e4**) P1: sleep EEG of the patient at the age of 4 yrs (longitudinal bipolar montage; sensitivity = 200 μV/cm, EMG1 = left deltoid muscle, PNG = pneumogram) is characterized by the absence of the typical electroencephalographic patterns of non-REM sleep and the presence of diffuse rapid activity and multifocal paroxysmal abnormalities. Dy: days of life; mth: months of life; Wgs: weeks of gestation; Yrs: years of life.

**Table 1 biomolecules-14-01626-t001:** Familial, molecular, clinical, and biochemical presentations of probands with *PIGW*-related condition.

Patient	Sex	Ancestry, Family History	*PIGW* (NM_178517.5) Variants, Status, and Inheritance	Presentation	Prenatal Findings	Birth Presentation	
P2	F	Danish, nc	c.50C>T, p.(Thr17Ile)/c.1202del, p.(Leu401GlnfsTer6); ht parents	PostN	Normal	37 + 5. Birth W 4040 g.	
[14]	F	Egyptian, 1st cousins	c.77C>T, p.(Leu26Ser), hz	PreN/PostN	Polyhydramnios, maternal diabetes	Sp delivery at term. W 3700 g (+0.98 SD), L 50 cm (+0.46 SD); APGAR 9–10	
P1	F	Italian, nc; previous miscarriage	c.106A>G, p.(Arg36Gly)pat/c.1227T>G, p.(Cys409Trp) mat	PreN/PostN	T1 increased risk, combined test for trisomy 21 (1/94); T2 (20 wgs): overriding aorta	Sp delivery 37 + 6 wgs. W 2.7 kg (90°pc); APGAR 9/9	
P3	F	France, previous miscarriage	c.106A>G, p.(Arg36Gly), h	PreN	Short long-bones with the femoral bowing, short hands and feet, cardiopathy and a Blake’s pouch cyst	TOP, 19 + 5 wgs, OFD < 3rd pc, W 50°pc, L 25°pc. Nuchal oedema	
P4	M	France, sib of P3	c.106A>G, p.(Arg36Gly), hz	PreN	Short long-bones, omphalocele, polydactyly	TOP 14 + 3 wgs	
[13] Family 9, II 1	M	NR	c.106A>G, p.(Arg36Gly), hz	PreN (fetus)	T2 (18 wgs): MCA, posterior fossa anomaly; TOP	NA	
[13] Family 9, II 2	F	NR	c.106A>G, p.(Arg36Gly), hz	PreN (fetus)	T1 (12 wgs): posterior fossa malformation, MCA; TOP	NA	
[15] fetus 3	F	Italian, nc	c.106A>G, p.(Arg36Gly), hz	PreN (fetus)	T1: increased NT (3.8 mm), SUA and omphalocele; T2 (15 wgs): brain malformations, dysmorphisms; short limbs; TOP at 17 wgs	NA	
[11]	M	Nc; elder brother: DD, death at 7 mth, RRI	c.178G>A,p.(Asp60Asn)pat/c.462A>T, p.(Arg154Ser) mat	PostN	NR	Sp delivery after 39 wgs. Pneumonia at 15 days of life	
[4]	F	Mexico, likely c (runs of hz at SNP-array)	c.199C>G, p.(Pro67Ala), hz	PostN	NR	NR	
[9]	NR	Japanese, nc	c.211A>C, p.(Thr71Pro)/c.499A>G, p.(Met167Val), comp htz	PostN	NR	NR	
[12] patient 1	M	1st cousins	c.460A>G, p.(Arg154Gly), hz	PostN	No	W 3.8 kg, L 52 cm, OFC 36 cm. Bradycardia, muscle hypotonia; APGAR 2/6/9	
[12] patient 2	F	2nd cousin of [9] patient 1	c.460A>G, p.(Arg154Gly), hz	PostN	No	W 3.5 kg, L 51 cm, OFC 35 cm. Hypotonia and a large tongue; APGAR 9/10/10	
[15] fetus 1	M	Italian, the couple’s grandparents were 1st cousins	c.827T>C, p.(Leu276Pro), hz	PreN (fetus)	T1: increased NT (4.10 mm, >95th pct); T2: pernasal thickness, polyhydramnios, dysmorphisms and MCA; TOP at 21 wgs	NA	
[15] fetus 2	F	Italian, the couple’s grandparents were 1st cousins, antecedent: [13] fetus 1	c.827T>C, p.(Leu276Pro), hz	PreN (fetus)	T1: low-risk combined test, normal NT; T2 (16 wgs): MCA; 18 wgs: increased pernasal thickness, dysmorphisms, and MCA; 20 wgs: brain anomalies; TOP at 21 wgs	NA	
[10]	F	Chinese, nc	17q12 deletion ~1.4 Mb pat/c.346 T>G, p.(Phe116Val) mat	PostN	NR	Sp delivery at term. W 3200 g; APGAR after resuscitation 7–8–10	
Statistics (%)	F 10/15 (66.7)	C 4/16 fam (25.0)|Dead sibling/miscarriage 3/16 (18.7)	Missense (12/14); frameshift (1/14); deletion (1/14)	PreN 9/16 (56.2); fetus 7/16 (43.7); PostN (43.7)	Fetal signs 9/16 (56.2)|T1 7/9 (77.7)	NA	
**Patient**	**Heart Anomalies**		**Brain Anomalies**	**Other Congenital Anomalies**	**Skeletal Anomalies**	**Extremities Anomalies**	
P2	Normal US at 1 and 2 yrs		Normal MRI	Bil hydronephrosis	NA	Normal	
[14]	Normal US		MRI: Bil enlargement of the cortical fronto-insular subarachnoid space and of the ventricles, thin CC and brainstem, white matter atrophy, moderate cerebellar atrophy	NR	Pectus excavatum, scoliosis	Long fingers with right fifth finger clinodactyly, and rocker bottom feet	
P1	US: vsd type ToF, altered pressures in the pulmonary artery, asd, ostium secundum type		Normal MRI	Right inguinal hernia	NR	Low-set left toe, overcrowding of toes	
P3	Overriding aorta		Microcephaly, enlarged ventricles and cerebellum hypoplasia.	Bil renal dysplasia	PME XR: shortened long bones, enlarged metaphysis, short ribs, and platyspondyly	Brachypahalangy	
P4	Asymmetry of the great arteries		NR	Bil renal dysplasia, omphalocele, anal atresia	PME XR: shortened long bones, platyspondyly	Right duplicated hallux	
[13] Family 9, II 1	NR		US (18 wgs): Dandy Walker malformation	US: hydronephrosis, genital hypoplasia	NR	NR	
[13] Family 9, II 2	NR		US (12 wgs): Dandy Walker malformation	US: diaphragmatic hernia, dysplastic kidneys, hydronephrosis	NR	NR	
[15] fetus 3	NR		US (15 wgs): third ventricle enlargement and cyst in the posterior fossa; MRI (16 wgs) dysmorphic and extra-rotated cerebellar vermis	US: SUA and a bowel-containing omphalocele; PME: bil renal ptosis and bicornuate uterus	US/PME: short limbs	PME: short halluces	
[11]	Normal US		MRI: widening of the subarachnoid space in both frontotemporal regions	Umbilical hernia and bilateral indirect inguinal hernias	Pectus excavatum	Bil mild flexion contractures of all fingers	
[4]	NR		NR	NR	Pectus excavatum	Bil mild flexion contractures of all fingers	
[9]	NR		NR	Inguinal hernia	NR	Normal	
[12] patient 1	NR		Normal MRI	Ankiloglossia	NR	Thumb-in-fist posture	
[12] patient 2	NR		Normal MRI	NR	NR	NR	
[15] fetus 1	NR		NR	PME: bil hydronephrosis	US/PME: short limbs	PME: bil brachydactyly with clinodactyly of the 5th finger, clubfeet	
[15] fetus 2	NR		US/MRI (20 wgs): enlargement of sub-arachnoid spaces, mild ventriculomegaly, cerebellar vermis hypoplasia, CC hypoplasia and dysmorphic features of brainstem with a suspected diencephalic–mesencephalic junction dysplasia	US/PME: hepatomegaly, bil hydronephrosis; PME: incomplete lung lobulation, intestinal malrotation, bicornuate uterus, and clitoris hypertrophy	US/PME: short limbs	US/PME: bil brachydactyly with clinodactyly of the 5th finger, clubfeet, short halluces	
[10]	Patent ductus arteriosus		MRI: cerebellar atrophy and bil enlargement of the cortical subarachnoid space and ventricles	NR	NR	NR	
Statistics (%)	4/16 (25.0)		8/16 (50.0)	12/16 (75.0)	8/16 (50.0)	10/16 (62.5)	
**Patient**	**Facial Dysmorphisms**		**Neurodevelopment**	**Seizure Onset**	**Seizure Semiology**	**EEG Abnormalities**	**Seizure Management**
P2	(photo) Brachycephaly, dolichocephaly, thin feathery hair, hypotelorism, deep-set eyes, large cup-formed ears, and flat nasal bridge		Global DD with delayed milestones, limited head control. Developmental regression (10 mth). Profound, global DD (no eye contact, no speech, 3 yrs)	10 mth	Tonic-spams, progressive epilepsy	EEG at 1 yr: overall hypsarrhythmia, characterized by cyclical variations during wakefulness and sleep. EEG during wakefulness showed periods of well-organized activity beyond 12 Hz, diffusely mixed with periods of theta activity without lateralization, alternating with periods of rhythmic delta activity mixed with sharp-waves, localized in multiple foci in both hemispheres. EEG during sleep: The lack of definitive sleep spindles and K-complexes, instead is seen a more persistent slow-wave pattern with 1.5–2 Hz activity, intermixed with sharp-waves as periodic discharges, where there may be asynchronous or synchronous activity in the two hemispheres. Ictal EEG at 3 yrs showed multifocal epileptiform changes.	Difficult to control: tried topiramat, levetiracetam, vigabatrin, pyrodoxin, clobazam, initiation of a ketogenic diet
[14]	(photo) low frontal hairline, the wide nasal bridge with a bulbous nasal tip and protruding columella, short philtrum, wide-spaced teeth, and the stem of the antihelix was fused to the helix		Global DD (sat unsupported at 12 mth; control of the trunk at 36 mth; unable to stand only with support from age 10 years), severe ID with absent speech	10 mth	Daily generalized tonic seizures with lip cyanosis since 10 mth; epileptic spasms/West syndrome since 13 mth; bilateral tonic seizure since 16 mth and focal-onset seizures; less than a seizure per year from 22 mth to 8 yrs; seizure free from 8 yrs to follow-up at 12 yrs	High-amplitude hypsarrhythmic pattern (13 mth); slowing of the background activity with single sharp-waves over the fronto-temporal bil regions during drowsiness (7 yrs)	Controlled with phenobarbital up to 13 mth; then treated with adenocorticotropic hormone (ACTH) with remission since 16 mth; then started Valproid acid without remission, Vigabatrin added at 20 mth
P1	(photo) squared/coarse face with large/prominent forehead. Anteverted, slightly low-set ears. Depressed nasal root, short nose, marked and protruding philtrum, slightly widened mouth with thin upper lip. Diastema. Protruding tongue. Micrognathia		Psychomotor DD, absent speech, immature gross and fine motor skills	No clinically recognizable epileptic episodes	NR	Atypical/disorganized, with unusually large amplitude, presence of continuous alpha band activity on the bilateral anterior areas and multifocal paroxysmal abnormalities (2 yrs 11 mth); EEG during sleep showed (a) the absence of the typical elements of physiological organization, while differentiating phases of drowsiness/sleepiness from phases of deeper sleep; (b) the presence of diffuse rapid activity that with falling asleep also increases in amplitude becoming dominant; (c) the presence of multifocal paroxysmal abnormalities (45 mth)	NR
P3	(photo) Brachycephaly, hypertelorism, low set ears		NA	NA	NA	NA	NA
P4	(photo) Cleft palate, turricephaly, long philtrum		NA	NA	NA	NA	NA
[13] Family 9, II 1	NR		NA	NA	NA	NA	NA
[13] Family 9, II 2	NR		NA	NA	NA	NA	NA
[15] fetus 3	(photo) US (15 wg): nasal bone hypoplasia, micrognathia; PME: brachycephaly, hypertelorism, anteverted nares, micrognathia, low-set ears		NA	NA	NA	NA	NA
**Patient**	**Facial Dysmorphisms**		**Neurodevelopment**	**Seizure Onset**	**Seizure Semiology**	**EEG Abnormalities**	**Seizure Management**
[11]	(reported) coarse facial features, wide nasal bridge, tent-shaped lips, high, narrow palatine arches		Severe psychomotor DD	70 dys	Partial-onset epileptic seizures	Sharp waves, sharp, slow discharges in the temporal and frontal regions	Treated by oxcarbazepine and anti-epileptic drugs but not be completely controlled even after temperature fall
[4]	(reported) anteverted nares, tented upper lip		Profound, global DD: inability to roll, poor head control, absence of tracking	7 mth	Infantile spasms	NR	NR
[9]	(reported) broad nasal bridge and tented upper lip		Profound DD	NR	Clusters of tonic spasms, West syndrome	High-amplitude hypsarrhythmic pattern	NR
[12] patient 1	NR		Psychomotor DD (sit at 11 mth, crawl at 13 mth, walk at 17 mth)	5 yrs	Seizure with atonia of the right leg and tendency to fall backwards	Multifocal spike-wave complexes especially while asleep	Improvement with sulthiame and clobazam therapy
[12] patient 2	(photo, partial; reported) macroglossia		Inactivity; Psychomotor DD (sit at 12 mth, crawl at 30 mth, not walking at 4 yrs)	2 yrs	First episode of atonic seizure followed by frequent seizures with myoclonic tongue movements, profuse salivation, fecal incontinence, and increased arm stiffness	Multifocal spike-wave complexes	Anti-convulsive treatment was only partially effective
[15] fetus 1	(photo) US: long philtrum; PME: dysmorphic ears, flat nose with anteverted nares, long philtrum, microretrognathia		NA	NA	NA	NA	NA
[15] fetus 2	(photo) US: nasal bone hypoplasia, micrognathia; PME: brachycephaly, flat glabella, anteverted nares, long and deep philtrum, cleft palate, retrognathia		NA	NA	NA	NA	NA
[10]	(reported) anteverted nares, tented upper lip, short philtrum, widely spaced teeth, and inner canthus		Psychomotor DD	9 mth	Generalized tonic-clonic seizures (during pneumonia at 9 mth; without fever at 15 mth); clusters of epileptic spasms/West syndrome (in the course of pneumonia at 23 mth); refractory epilepsy; Recurrent epileptic spasms with muscular atrophy and hypotonia, asymmetric tonic seizures worsening during fever and infections	High-amplitude hypsarrhythmic pattern on an interictal electroencephalogram (23 mth)	Not controlled by poly-drug therapy (6 drugs) and vit B6 supplementation
Statistics (%)	12/16 (75.0)		9/9 (100)	(70 dys–5 yrs); Me 44.1 mth	8/9 (88.9)	9/9 (100)	Partial control or uncontrolled epilepsy (5/8, 62.5)
**Patient**	**Other Neurologic Findings**		**Behavior**	**Infections**	**Other Clinical Findings**	**ALP Measurement**	**Other Biochemical Findings**
P2	Mild axial hypotonia, difficulties swallowing		At 3 yrs, she had no eye contact and no speech	Frequent respiratory infections, annually 5–10 contacts to hospital. Pneumonia leading to lethal asystole at 3 yrs 10 mth	Height and weight > 3 SD. Hypermetropia, with +5.75 bilat	High (610–940 U/L), measured 12 times from 9 to 42	NA
[14]	Global hypotonia, intermittent hand stereotypies (hand washing and mouthing); limited, wide-based walking; ataxic gait		Limited social interaction, poor eye contact	RRI	Three congenital hemangiomas. Deficiency of the left central auditory pathway. Horizontal nystagmus	High at two measurements (425 and 380 U/L)	Normal metabolic tests and transferrin isoelectrofocusing
P1	NR		Absent speech, poor eye engagement, limited pointing	NR	Tooth agenesis.	High (519 U/L) at 3 yrs	Reduced CD16 at MFI from neutrophils
P3	NA		NA	NA	NR	NA	NA
P4	NA		NA	NA	NR	NA	NA
[13] Family 9, II 1	NA		NA	NA	NR	NA	NA
[13] Family 9, II 2	NA		NA	NA	NR	NA	NA
[15] fetus 3	NA		NA	NA	NR	NA	NA
[11]	No		NR	RRI	Failure to thrive, poor weight gain. Bil lung pneumonia requiring intubation	High (414–798 U/L)	Normal levels of immunoglobulins and lymphocyte subsets; normal cerebrospinal fluid analysis
[4]	NR		NR	Frequent aspiration pneumonia.	Failure to thrive. Poor tracking, vertical nystagmus	High (473 U/L and 495 U/L) pre-diagnosis, ranging from 235 to 575 U/L at 9–11 mth	FC: decreased anchor protein expression (82% granulocytes; 81% lymphocytes). Cluster differentiation: decreased CD16 and CD24 cell surface expression (>60% granulocytes), CD14 expression (88% on monocytes), and CD59 expression (42% lymphocytes)
[9]	NR		NR	NR	No	High (2000 U/L)	FC: Reduced levels of the GPI-Aps. Reduced CD59 and FLAER on lymphocytes, reduced CD14 on monocytes. Normal routine laboratory investigations
[12] patient 1	Neonatal hypotonia		Absent speech (7 yrs), autistic traits	NR	Ankyloglossia	Up to 230 U/L (normal)	Low MFI for CD16 (2832); reduced CD16 (7214)
[12] patient 2	Neonatal hypotonia		NR	NR	Abnormal brainstem auditory evoked response	Up to 305 U/L (normal)	Low MFI for CD16 (3070); reduced CD16 (7439)
[15] fetus 1	NA		NA	NA	NR	NA	NA
[15] fetus 2	NA		NA	NA	NR	NA	NA
[10]	Severe ID. Hypotonia. Muscular atrophy		NR	RRI	Corneal pannus, keratitis, and conjunctivitis	High (552–995 U/L)	NR
Total (%)	6/9 (88.9)		4/9 (44.4)	5/9 (55.6)	8/16 (50.0)	High 7/9 (77.8)	FC: 5/6 (83.3)

Familial, molecular and clinical presentations of probands with *PIGW*-related condition. asd: atrial septal defect; bil: bilateral; c: consanguineous couple; CC: corpus callosum; comp htz: compound heterozygous; DD: developmental delay; dys: days of life; F: female; FC: Flow-cytometry; GA: gestational age; hz: homozygous; gws: gestational weeks; ID: intellectual disability; M: male; mat: maternal inheritance; MCA: multiple congenital anomalies; Me: arithmetic media; MFI: mean fluorescence intensities; mth: months of age; NA: not applicable; nc: non consanguineous couple; NR: not reported; pct: percentile; pat: paternal inheritance; PreN: pre-natal; PME: post-mortem examination, including autopsy; PostN: post-natal; RRI: recurrent respiratory infections; SNV: single nucleotide variant; Sp: spontaneous; SUA: single umbilical artery; T: trimester; ToF: tetralogy of Fallot; vsd: ventricular septal defect; yrs: years of age; wgs: weeks of gestation.

## Data Availability

The data presented in this study are available upon request from the corresponding author. Due to privacy restrictions, they are not publicly available.

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
