# Peer review of "In-Depth Phenotyping of PIGW-Related Disease and Its Role in 17q12 Genomic Disorder"

_biomolecules, 2024, doi:10.3390/biom14121626_

Round 1

Reviewer 1 Report

Comments and Suggestions for Authors

The authors present an interesting study on PIGW-related disease and the possible role of this gene in genomic pathologies caused by imbalances in the 17q12 region. These pathologies are extremely rare, with very few cases described, so any contribution is important to improve our knowledge in the field. For this reason, the clinical description of the patients studied must be exact and detailed, and indeed the authors promise ‘In-depth phenotyping’ in the title of the manuscript. Unfortunately, the description of certain dysmorphological features of the reported patients is not supported by measurements and, above all, is not confirmed by the images presented. Dysmorphological assessments must always be made according to precise and objective criteria and not based on personal impressions. In this regard, it may be useful to follow the indications provided by specific texts, such as that of J Hall, J Allanson, K Gripp, and A Slavotinek: Handbook of physical measurements - Oxford University press, and scientific articles, such as PMID 19127575, 24124000, 20920337, and others.

In some places, the text denotes a perhaps too literal translation from another language, so it would benefit from revision by a native English speaker familiar with scientific terms, particularly medical ones.

I also have a few other observations, which I reproduce below.

Page 3, line 99: "100 mcl" should be "100 µL".

Page 3, lines 121-122: Can the authors explain why a prenatal diagnosis was not performed following a prenatal screening with increased risk for trisomy 21? Was it the couple's choice or a lack of information from those who carried out the test? We would welcome a comment from the authors on this point.

Page 3, lines 125-126: although widely used in the scientific literature, the term 'chromosomal microarray' is incorrect and should be avoided. This test should be better defined as a "genomic microarray (GMA)" as it provides information on the possible presence of genomic imbalances without telling us anything about the chromosomal structure.

Page 4, line 147: the authors write "She had low muscle masses but normal strength". How was muscle strength measured?

Page 4, lines 155-156: the authors write ‘Surgical treatment for correction of the right inguinal hernia was performed’ but this feature was not mentioned in the previous description of the patient.

Page 4, line 159: the authors write 'increased plasma values of ALP (519U/L)'. At what age was the assay performed? Are the values really high?

Page 4, line 165: "dolichocephaly hypotelorism" should be "dolichocephaly, hypotelorism".

Page 5, lines 189-190: what were the ALP values and at what age were they measured?

Page 5, line 198: Instead of 'incurvated humerus', perhaps 'curved humerus' or 'bent humerus' would be better. Moreover, this aspect was not confirmed by the post-abortion x-ray. It is also curious that certain features detected after abortion were not seen on ultrasound (microcephaly, bilateral renal dysplasia, overriding aorta and small brain with enlarged ventricles and cerebellum hypoplasia). A comment from the authors on this point would be welcome.

Page 5, line 199: perhaps "severe prognosis" would be better than "severe prognostic".

Page 5, line 209: the authors report that the foetus had an omphalocele. The image is not very clear, but it seems more likely that it was gastroschisis.

Page 8, line 245: the authors write "widened mouth". How was this aspect assessed? The description of the ears also does not correspond to what is shown in the image. The same considerations apply to the description of patient P2.

Page 19, line 362: instead of "incurvated humerus" perhaps "curved humerus" or "bent humerus" would be better.

Page 19, lines 374-375: some features indicated by the authors as common are doubtful, or at least not well evident in the photographic documentation (widened, down slanting mouth with thin upper lip, and low set, anteverted, cupped ears).

Page 20, lines 401-402: This aspect, which could be important, should be better specified. The values reported for P1 do not seem really high, those for P2 are not reported at all.

Page 20, line 433: probably ‘diaphragm and wall’ should be ‘diaphragm and abdominal wall’.

Page 20, line 469: the use of the double round bracket: (c.1227T>G, p.(Cys409Trp)) may be confusing, perhaps it would be better to write [c.1227T>G, p.(Cys409Trp)].

Page 21, line 435: in the sentence 'there are no prenatal diagnosis of PIGW during pregnancy' there is an unnecessary repetition (prenatal - during pregnancy). In any case, "there are no prenatal diagnosis" should be "there is no prenatal diagnosis".

Page 21, lines 475-479: the presence of high ALP values in P1 must be confirmed.

Page 23, line 530: probably ‘wall defects’ should be ‘abdominal wall defects’.

Page 23, line 540: P4 should be P2.

Page 24, line 595: The term 'chromosomal microarray', although widely used in the scientific literature, is incorrect and should be avoided. This test should be better defined as a "genomic microarray (GMA)" as it provides information on the possible presence of genomic imbalances without telling us anything about the chromosomal structure.

Overall, the manuscript is interesting and deserves to be published, but it needs some modifications. Given the small number of patients studied, moreover, conclusions should be very cautious and based on particularly careful observation of the patients' clinical characteristics.

Comments on the Quality of English Language

In some places, the text denotes a perhaps too literal translation from another language, so it would benefit from revision by a native English speaker familiar with scientific terms, particularly medical ones.

Author Response

Dear Reviewer

We are very grateful to the Reviewer for the constructive comments on our manuscript. We have implemented the Reviewer’s comments and suggestions and wish to submit a revised version of the manuscript for further consideration. 

Below, we provide a point-by-point response explaining how we have addressed each of the Reviewers’ comments.

In the attached file responses are marked in red. 

Response to Reviewer 1 Comments

Summary

The authors present an interesting study on PIGW-related disease and the possible role of this gene in genomic pathologies caused by imbalances in the 17q12 region. These pathologies are extremely rare, with very few cases described, so any contribution is important to improve our knowledge in the field. For this reason, the clinical description of the patients studied must be exact and detailed, and indeed the authors promise ‘In-depth phenotyping’ in the title of the manuscript. Unfortunately, the description of certain dysmorphological features of the reported patients is not supported by measurements and, above all, is not confirmed by the images presented. Dysmorphological assessments must always be made according to precise and objective criteria and not based on personal impressions. In this regard, it may be useful to follow the indications provided by specific texts, such as that of J Hall, J Allanson, K Gripp, and A Slavotinek: Handbook of physical measurements - Oxford University press, and scientific articles, such as PMID 19127575, 24124000, 20920337, and others.

In some places, the text denotes a perhaps too literal translation from another language, so it would benefit from revision by a native English speaker familiar with scientific terms, particularly medical ones.

I also have a few other observations, which I reproduce below.

Overall, the manuscript is interesting and deserves to be published, but it needs some modifications. Given the small number of patients studied, moreover, conclusions should be very cautious and based on particularly careful observation of the patients' clinical characteristics.

Point-by-point response to Comments and Suggestions for Authors

Comment 1: Page 3, line 99: "100 mcl" should be "100 µL".

Response 1: Thank you for the comment. We have changed according to the suggestion (page 3, line 97)

Comment 2: Page 3, lines 121-122: Can the authors explain why a prenatal diagnosis was not performed following a prenatal screening with increased risk for trisomy 21? Was it the couple's choice or a lack of information from those who carried out the test? We would welcome a comment from the authors on this point.

Response 2: Thank you for pointing this out. The couple were counseled during the first trimester in another center, so we cannot provide the details of the counselling. However, in the light of a concise and focused clinical report, we have described the prenatal analysis and results in lines 122-124: "After genetic counselling, we performed amniocentesis with inconclusive karyotype, FISH for 22q11.2 deletion and genomic microarray". We believe that this description is sufficient for our purposes.

Comment 3: Page 3, lines 125-126: although widely used in the scientific literature, the term 'chromosomal microarray' is incorrect and should be avoided. This test should be better defined as a "genomic microarray (GMA)" as it provides information on the possible presence of genomic imbalances without telling us anything about the chromosomal structure.

Response 3: Thank you for your suggestion. We agree with adopting “genomic microarray” instead of “chromosomal microarray” in the manuscript. The changes have been incorporated into the manuscript (lines 124 and 613).

Comment 4: Page 4, line 147: the authors write "She had low muscle masses but normal strength". How was muscle strength measured?

Response 4: The patient was examined by expert neurologists and paediatricians, who conducted objective assessments. We have modified the text as follows: “During objective neurologic examination, she had low muscle masses but normal strength” (page 4, lines 145-147).

Comment 5: Page 4, lines 155-156: the authors write ‘Surgical treatment for correction of the right inguinal hernia was performed’ but this feature was not mentioned in the previous description of the patient.

Response 5: Thank you for pointing this out. We agree with this comment. Therefore, we added a sentence in order to make this point clearer “We also noted unilateral inguinal hernia” (page 4, lines 143-144).

Comment 6: Page 4, line 159: the authors write 'increased plasma values of ALP (519U/L)'. At what age was the assay performed? Are the values really high?

Response 6: Thank you for your comment. We agree that it is a crucial aspect for the description and discussion of the phenotype. We implemented the text with the requested information concerning the age at which the assay was performed (page 4, line 159). For clarity, we have included the reference values in the manuscript. The ALP value in P1 was 519 U/L measured at 3 years of age, which is elevated according to the range values provided by the laboratory for age (156-369 U/L).

Comment 7: Page 4, line 165: "dolichocephaly hypotelorism" should be "dolichocephaly, hypotelorism".

Response 7: Thank you for drawing this to our attention. We have modified the item (page 4, line 165).

Comment 8: Page 5, lines 189-190: what were the ALP values and at what age were they measured?

Response 8: Thank you for your comment. We agree that this is an important point for phenotype description and discussion. Please note that ALP values for P2 are reported in table 1 “High (610-940 U/L)”. ALP was measured 12 times from 9 to 42 months of age and it was always elevated between 610-940 U/L (reference 134-518U/L). For clarity, we have added the requested information in the text (page 5, lines 189-191)

Comment 9: Page 5, line 198: Instead of 'incurvated humerus', perhaps 'curved humerus' or 'bent humerus' would be better. Moreover, this aspect was not confirmed by the post-abortion x-ray. It is also curious that certain features detected after abortion were not seen on ultrasound (microcephaly, bilateral renal dysplasia, overriding aorta and small brain with enlarged ventricles and cerebellum hypoplasia). A comment from the authors on this point would be welcome.

Response 9: Thank you for this comment. We propose to adopt “femoral bowing HP:0002980”; according to Human Phenotype Ontology term (page 5, line 200; Table 1).

As the Reviewer pointed out, certain features detected through the fetal pathological examination after abortion (post-mortem) were not seen on prenatal ultrasound examination. This might be due to various factors depending on accessibility, sensitivity, and specificity of the procedure and on the gestational age.

A fetal pathological examination offers the possibility of a comprehensive assessment and description of the fetal phenotype through external, dysmorphology examination, direct measurements, and visceral examination (macroscopic and histologic). The process may result in the objectivation of signs that have not been observed on ultrasound examination. A number of studies in the literature have highlighted discrepancies between prenatal sonographic assessment and post-mortem examination and autopsy, particularly for fetuses before 20 weeks of gestation (PMID: 36260644), for dysmorphology and minor external anomalies, and certain organs (PMID: 22122607, PMID: 20336773, PMID: 34968018).

In this study, fetal pathological examination has been performed according to the French national protocol for both fetuses, in which the termination of pregnancy was done at different gestational ages

(https://www.has-sante.fr/upload/docs/application/pdf/2014-06/protocole_examen_autopsique_foetal_neonatal_cd_20140528_vd.pdf). 

Comment 10: Page 5, line 199: perhaps "severe prognosis" would be better than "severe prognostic".

Response 10: Thank you for your comment. We have made the change you suggested (page 5, line 201).

Comment 11: Page 5, line 209: the authors report that the foetus had an omphalocele. The image is not very clear, but it seems more likely that it was gastroschisis.

Response 11: Thank you for the comment. On the left profile, you can see a membrane around the viscera, confirming that it is an omphalocele.

Comment 12: Page 8, line 245: the authors write "widened mouth". How was this aspect assessed? The description of the ears also does not correspond to what is shown in the image. The same considerations apply to the description of patient P2.

Response 12: The patient was evaluated by external examination by a team of clinical geneticists and paediatricians using standard methodology. "Widened mouth" corresponds to the patient's tendency to keep the mouth open for most of the time during the medical examination.  Furthermore, this tendency was also corroborated by parental reports.

Comment 13: Page 19, line 362: instead of "incurvated humerus" perhaps "curved humerus" or "bent humerus" would be better.

Response 13: Thank you for pointing this out. We have adopted the term "femoral bowing HP:0002980"; according to the Human Phenotype Ontology (line 368).

Comment 14: Page 19, lines 374-375: some features indicated by the authors as common are doubtful, or at least not well evident in the photographic documentation (widened, down slanting mouth with thin upper lip, and low set, anteverted, cupped ears).

Response 14: Thank you for your comment. A team of expert clinical geneticists conducted external examinations of the patients and provided a clinical evaluation. Given the limited number of PIGW patients and the paucity of photographic documentation of the facial phenotype, we proposed to show and compare facial pictures of unrelated P1 and P2 at different ages. The collection and analysis of dysmorphological data enabled the identification of individual features and the discovery of similarities, particularly concerning the mouth, philtrum, and nose. We rephrase the sentence as follows: "We also first document the evolution of phenotype in P1 and P2, with slight similarities concerning the facial appearance, especially concerning the nose (depressed nasal root),the philtrum (marked philtrum), and the mouth (widened, down slanting, thin upper lip) (lines 384-387).

Comment 15: Page 20, lines 401-402: This aspect, which could be important, should be better specified. The values reported for P1 do not seem really high, those for P2 are not reported at all.

Response 15: Thank you for your consideration. We agree that this is an important point for phenotype description and discussion. The ALP value for P1 was measured at the age of 3 years (519 U/L, reference 156 - 369 U/L). Please note that the ALP value for P2 has been included in Table 1. For clarity, we have included the text with this information (page 5, lines 189-191)

Comment 16: Page 20, line 433: probably ‘diaphragm and wall’ should be ‘diaphragm and abdominal wall’.

Response 16: Thank you for the suggestion. We have made the change you suggested (page 20, line 424).

Comment 17: Page 20, line 469: the use of the double round bracket: (c.1227T>G, p.(Cys409Trp)) may be confusing, perhaps it would be better to write [c.1227T>G, p.(Cys409Trp)].

Response 17: Thank you for your comment. We have made the change you suggested (page 20, line 489).

Comment 18: Page 21, line 435: in the sentence 'there are no prenatal diagnosis of PIGW during pregnancy' there is an unnecessary repetition (prenatal - during pregnancy). In any case, "there are no prenatal diagnosis" should be "there is no prenatal diagnosis".

Response 18: Thank you for the suggestion. We have made the change you suggested (page 21, line 451-452).

Comment 19: Page 21, lines 475-479: the presence of high ALP values in P1 must be confirmed.

Response 19: Thank you for your comment. Please see our response to comment 8.

Comment 20: Page 23, line 530: probably ‘wall defects’ should be ‘abdominal wall defects’.

Response 20: Thank you for your comment. We have made the change you suggested (line 548).

Comment 21: Page 23, line 540: P4 should be P2.

Response 21: Thank you for pointing this out. We have corrected “P4” with “P2” (page 23, line 558).

Comment 22: Page 24, line 595: The term 'chromosomal microarray', although widely used in the scientific literature, is incorrect and should be avoided. This test should be better defined as a "genomic microarray (GMA)" as it provides information on the possible presence of genomic imbalances without telling us anything about the chromosomal structure.

Response 22: Thank you for your suggestion. We agree. We have therefore changed "chromosomal microarray" to "genomic microarray" in the manuscript (lines 126 and 613).

General Comments:

General comment 1: For this reason, the clinical description of the patients studied must be exact and detailed, and indeed the authors promise ‘In-depth phenotyping’ in the title of the manuscript. Unfortunately, the description of certain dysmorphological features of the reported patients is not supported by measurements and, above all, is not confirmed by the images presented. Dysmorphological assessments must always be made according to precise and objective criteria and not based on personal impressions. In this regard, it may be useful to follow the indications provided by specific texts, such as that of J Hall, J Allanson, K Gripp, and A Slavotinek: Handbook of physical measurements - Oxford University press, and scientific articles, such as PMID 19127575, 24124000, 20920337, and others.

Response to General comment 1: Thank you for this comment. We agree with the Reviewer that it is extremely important to evaluate all clinical features that could be part of a broad phenotype due to PIGW variants. As the Reviewer points out, this is even more critical in an ultra-rare condition where only a limited number of patients have been described, including some fetuses. In this regard, a team of experienced clinical geneticists has evaluated the dysmorphological features of all the patients (children and fetuses) at the various Institutions according to precise and objective criteria. In the study, we assessed deep-phenotyping of all the patients, reporting detailed clinical (external examination, imaging, and eventual autopsy) and biochemical presentation. The authors provided comprehensive descriptions of the patients, which are also summarized in Table 1. Additionally, illustrative images of the most salient features in each patient are presented in Figure 1. The relatively small number of PIGW patients (mostly reported through written descriptions and rarely shown by published photographs) currently precludes the definition of a specific facial appearance associated with this condition. In view of these limitations, we proposed a comparison of facial pictures of P1 and P2 at different ages, and of P3 and P4 at the time of fetal pathological examination.  The collection and analysis of dysmorphological data enabled us to highlight individual features and to identify some similarities, especially comparing unrelated patients (P1, P2) who were evaluated over time.

General Comment 2: In some places, the text denotes a perhaps too literal translation from another language, so it would benefit from revision by a native English speaker familiar with scientific terms, particularly medical ones.

Response to General comment 2: The manuscript has undergone professional extensive English revision.

Additional integrations by the authors:

  • Please note that we have made some minor integrations in Table 1, column “Facial dysmorphisms”, indicating how data have been reported (photo/reported) for each patient.
  • References in Table 1 have been corrected in the revised version of the manuscript.
  • Reference n 18 has been added.

Reviewer 2 Report

Comments and Suggestions for Authors

See attached Word document.

Comments on the Quality of English Language

See this topic already covered in the attached Word document.

Author Response

Dear Reviewer

We are very grateful to the Reviewer for the constructive comments on our manuscript. We have implemented the Reviewer’s comments and suggestions and wish to submit a revised version of the manuscript for further consideration. 

Below, we provide a point-by-point response explaining how we have addressed each of the Reviewers’ comments.

In the attached file responses are marked in red.

Response to Reviewer 2 Comments

Summary

Feresin, et al., have described four new cases of an ultra-rare disorder, resulting from variants in the PIGW gene. Only 12 other patients had previously been described, bringing the total now to 16. The authors describe their four new cases clinically, pathologically, and genetically (via whole-exome sequencing). Next, the authors collate all published cases into a massive table describing (where available) the clinical, molecular, biochemical, and familial history profiles of each of the 16 patients. This is followed by a discussion of the cohesive phenotypic presentation, together with the spectrum of all phenotypes that have been observed and reported. Taken together, these four additional case reports contribute to the knowledge of rare diseases; however, the manuscript requires significant additional attention.

Point-by-point response to Comments and Suggestions for Authors

Comment 1: Line 86: informed consent, not informed “consents”

Response 1: Thank you for your comment. We have made the change you suggested (line 84)

Comment 2: Line 169: At four days old, not “Four days old,”

Response 2: Thank you for your comment. We have made the change you suggested (line 169)

Comment 3: Line 194: The patient passed away, at 3 years and 10 months of age, due to ..., not “The patient passed away, 3 years and ten months old, due to…”

Response 3: Thank you for your comment. We have made the change you suggested (lines 195-196)

Comment 4: Line 197: couple from France, who had one previous unexplained miscarriage, not “…couple from France, who had previously an unexplained miscarriage.”

Response 4: Thank you for your comment. We have made the change you suggested (lines 198-199)

Comment 5: Line 207: The parents asked for, not “Parents asked for…”

Response 5: Thank you for your comment. We have made the change you suggested (line 210)

Comments 6: Lines 304-307: Both of these sentences are awkward and need to be rephrased; as they are written right now, it is impossible to deduce the meaning.

Response 6: Thank you for your advice. The manuscript has undergone professional extensive English revision. The sentences have been rephrased.

Comment 7: Line 337: …we analyzed, not “we then analyzed”

Response 7: Thank you for your comment. We have made the change you suggested (line 343)

Comments 8: Line 342: and great artery malformations, not “and great arteries malformation”

Response 8: Thank you for your comment. We have made the change you suggested (lines 348-249)

Comment 9: Line 364-366: “Although additional evidence is needed, overgrowth and tooth agenesis have already been mentioned in at least one patient with a PIGA pathogenic variant, …” not “Although additional evidence are needed, overgrowth and tooth agenesis have been already mentioned at least in one patient with PIGA pathogenic variant,”

Response 9: Thank you for your comment. We have made the change you suggested (lines 373-375)

Comments 10: Line 377: However, she presented with global developmental delay, which is a key finding in all PIGW children, and with behavioral anomalies, which have been less consistently reported.” (Not: However, she presented global developmental delay, with is a key finding in all PIGW children and behavioral anomalies, which have been more inconstantly reported.”)

Response 10: Thank you for your comment. We have made the change you suggested (line 391-393)

Comments 11: Line 450: “her second affected cousin” – do you mean her affected [second cousin]? Or that there was a first and there is now a second [cousin] (presumably a first cousin?) affected with the same condition? With the way the English here is written, the meaning is not clear.

Response 11: Thank you for your comment. We have made the change you suggested (lines 466-467)

Comments 12: Line 505: What is “Sud” America? Do you mean South America?

Response 12: Thank you for your comment. We have made the correction (line 524)

Comment 13: Line 527: associated with different manifestations. (“Associated to” is not proper English.)

Response 13: Thank you for bringing this to our attention. The requisite correction has been implemented (line 539)

Comments 14: Line 598: broad, continuous spectrum (not: broad, continuum spectrum).

Response 14: Thank you for your comment. We have made the change you suggested (line 616-617)

Comment 15: The sample size is very small. This is, of course, due to the nature of reporting on an ultra-rare disease. Nevertheless, it would be beneficial to add a sentence or two to the discussion on this point as a limitation to the findings in this study. Even with sixteen patients, it is doubtful that the true phenotypic spectrum has been fully described (because of the low case number). The phenotypic spectrum will likely only be expanded in the future as additional cases are identified. This paper brings the field an important step closer.

Response 15: Thank you for this comment. We agree with the assessment of the reviewer that the number of cases described to date is limited and that the phenotypic spectrum is likely to be expanded in the future if new cases are identified. In this regard, we strongly believe that in-depth phenotyping and reverse phenotyping are fundamental to clarify whether apparently isolated features actually represent an expansion of the specific clinical presentation. We agree with the reviewer that it is noteworthy to mention this in the discussion. Consequently, we have addressed this issue in paragraph 4.2 (lines 471-473). Furthermore, to emphasize this point, we have added the term 'ultra-rare' in the conclusion (line 616).

Comments 16 • The Methods – even the Supplemental Methods – do not adequately describe the variant elimination pipeline. How many variants were left in each patient at each step of your pipeline? How many compared to the reference genome? How many after MAF filtering? Etc. It would be helpful to provide a step-wise variant count – one table with a column for each patient would suffice – culminating in the 1 or more variants that were left for consideration. Right now, there is nothing to convince me that the PIGW variants were the ONLY possible variants (predicted to have high impact, and so forth) present in each of the patients; were there any other variants that could be disease-causing? What is the rationale to skip straight to the PIGW variants at the expense of others (what rationale was used to ignore other potentially high impact variants)? This ties into the statement in lines 216-217 that there were no other pathogenic variants that could explain the phenotype…there needs to be better evidence to back this up, rather than just asking the reader to take your word for it. Even something as simple as Table 1 in this paper (https://www.mdpi.com/2073-4425/13/2/334), for each of the four cases described herein, would go a long way toward clarification of this point.

Response 16: Thank you for bringing this issue to our attention. In accordance with the recommendations of the reviewer, the filtering and prioritisation processes have been presented in comprehensive detail in the Supplementary Information. It should be additionally noted that the pipelines utilised in the three accredited diagnostic laboratories are well-established, published, and detailed in the referenced publications listed below, which are included as references in the current manuscript.

  • Musante L, Faletra F, Meier K, Tomoum H, Najarzadeh Torbati P, Blair E, North S, Gärtner J, Diegmann S, Beiraghi Toosi M, Ashrafzadeh F, Ghayoor Karimiani E, Murphy D, Murru FM, Zanus C, Magnolato A, La Bianca M, Feresin A, Girotto G, Gasparini P, Costa P, Carrozzi M. TTC5 syndrome: Clinical and molecular spectrum of a severe and recognizable condition. Am J Med Genet A. 2022 Sep;188(9):2652-2665. doi: 10.1002/ajmg.a.62852. Epub 2022 Jun 7. Erratum in: Am J Med Genet A. 2023 Mar;191(3):910. doi: 10.1002/ajmg.a.63091. PMID: 35670379; PMCID: PMC9541101
  • Bayat A, Fenger CD, Techlo TR, Højte AF, Nørgaard I, Hansen TF, Rubboli G, Møller RS, Group DCCRS. Impact of Genetic Testing on Therapeutic Decision-Making in Childhood-Onset Epilepsies-a Study in a Tertiary Epilepsy Center. Neurotherapeutics. 2022 Jul;19(4):1353-1367. doi: 10.1007/s13311-022-01264-1. Epub 2022 Jun 20. PMID: 35723786; PMCID: PMC9587146.
  • Lefebvre M, Bruel AL, Tisserant E, Bourgon N, Duffourd Y, Collardeau-Frachon S, Attie-Bitach T, Kuentz P, Assoum M, Schaefer E, El Chehadeh S, Antal MC, Kremer V, Girard-Lemaitre F, Mandel JL, Lehalle D, Nambot S, Jean-Marçais N, Houcinat N, Moutton S, Marle N, Lambert L, Jonveaux P, Foliguet B, Mazutti JP, Gaillard D, Alanio E, Poirisier C, Lebre AS, Aubert-Lenoir M, Arbez-Gindre F, Odent S, Quélin C, Loget P, Fradin M, Willems M, Bigi N, Perez MJ, Blesson S, Francannet C, Beaufrere AM, Patrier-Sallebert S, Guerrot AM, Goldenberg A, Brehin AC, Lespinasse J, Touraine R, Capri Y, Saint-Frison MH, Laurent N, Philippe C, Tran Mau-Them F, Thevenon J, Faivre L, Thauvin-Robinet C, Vitobello A. Genotype-first in a cohort of 95 fetuses with multiple congenital abnormalities: when exome sequencing reveals unexpected fetal phenotype-genotype correlations. J Med Genet. 2021 Jun;58(6):400-413. doi: 10.1136/jmedgenet-2020-106867. Epub 2020 Jul 30. PMID: 32732226.

In order to provide further clarification and to facilitate understanding of the procedure, we are pleased to present a brief overview of the key steps involved.

  • Variants were annotated and prioritized based on the American College of Medical Genetics and Genomics (ACMG) guidelines.
  • Different possible modes of inheritance (sporadic de novo, dominant, recessive, X-linked) were considered.
  • A minor allele frequency (MAF) cut-off of ≤0.01% was utilized.
  • SNVs and INDELs were filtered by referring to public databases to rule out variants previously reported as polymorphism.
  • The variant’s pathogenicity was assessed using several in silico prediction tools.
  • The variant’s pathogenicity was also assessed with consultation of the ClinVar, the Human Gene Mutation Database professional (HGMD), Online Mendelian Inheritance in Man (OMIM) and DECIPHER.
  • Importantly, the variant's prioritization was the result of combining different strategies which include the diagnostic interpretation of WES data in the context of clinical and phenotypic data at multidisciplinary meetings, by systematic bibliographic review and public database consultation.
  • Finally, the selected variant/s were validated and familial segregation was performed by Sanger sequencing.

In accordance with the aforementioned procedure, we were able to identify independently, biallelic pathogenic or likely pathogenic PIGW variants, including the p.(Arg36Gly), which was previously reported as the molecular cause in three fetuses from two unrelated families (see Table 1 and references herein). Furthermore, our pipelines did not identify any other variants, classified as pathogenic, likely pathogenic, or variants of unknown significance, that could explain the observed phenotype in the probands, even when considering individual peculiar features. It is notable that the clinical presentation of our patients is consistent with that described in the literature. Furthermore, the deep phenotyping enabled the identification of new clinical features, including cardiac anomalies, which appear to represent a part of a broad phenotype, expanding the antenatal, post-mortem, and postnatal clinical spectrum of the PIGW-related disorder.

Comments 17: Along this same topic – because WES and not WGS was carried out, there is still the possibility that one or more variants that could still be pathogenic have been missed due to the methodology selected (most of the genome was not interrogated). This should be mentioned as a limitation in the Discussion.

Response 17: The recent guidelines of the American College of Medical Genetics and Genomics (ACMG) recommend the use of genome‐wide analyses, such as whole exome sequencing, at the beginning of the diagnostic approach in the care of pediatric patients presenting with one or more congenital anomalies (CA), developmental delay (DD) or intellectual disability (ID) (PMID: 34211152). Whole exome sequencing is currently a widely available first-tier examination in clinical practice, whereas whole genome sequencing is  the first test of choice  for only a small number of clinics and it is often performed in a research setting.

We identified by WES likely pathogenic and pathogenic variants in PIGW, which explain the clinical presentation of our patients. However, we cannot exclude the possibility that individual peculiar clinical features could be explained by one or more variants located outside the investigated genomic region. Nevertheless, in agreement with the Reviewer (comment 15), it is also plausible that individual clinical features might represent a part of the phenotypic spectrum which will likely only be expanded in the future as additional cases are identified.

Comment 18: Line 540: You refer here to P4. But I think you mean P2? Earlier in the paper, it was P2 that was put on a ketogenic diet (and with P4 being an aborted pregnancy, this also does not match up).

Response 18: Thank you for bringing this inaccuracy to our attention. We have made the necessary correction (line 558).

Minor Considerations

Comments 19• The abbreviation WES is first used in line 81, it should be defined here. And therefore it does not need to be defined again in line 90.

Response 19: Thank you for your comment. We have made the change you suggested (lines 79 and 88)

Comments 20: In multiple locations – line 91, and also the first line of the Methods in the Supplemental data for both patient 1 and patient 2 – it says that DNA was isolated from lymphocytes. Did you really cell sort out other white blood cells (neutrophils, etc.) in the blood so that you were only extracting DNA from lymphocytes? If not, then I think in both places the word you want is “leukocytes” not lymphocytes.

Response 20: Thank you for the comment. We apologize for the inaccuracy and we have made the appropriate change in the methods.

Comment 21: Line 101: Does the ‘ symbol here mean minutes? If so, just write minutes, please. Or is it a typo?

Response 21: Thank you for bringing this inaccuracy to our attention. We apologize for the error, which was caused by a typographical mistake. We have made the change you suggested (line 99)

Comment 22: Legend for Figure 3: Right now, MCA is defined after panel (d), but would be better placed earlier, after panel (b), where the MCA abbreviation is actually used.

Response 22: Thank you for the comment. We have made the change you suggested (lins 319, 322)

Comments 23: Line 443: The abbreviation ES has not been defined yet?

Response 23: Thank you for the comment. We have made the change (line 459)

Major Considerations

Major Considerations: • Throughout the entire manuscript, the English is not quite right, which makes this manuscript tough to read. There are sentence structure problems, there are punctuation problems, along with many preposition problems (prepositions where they should not be, no preposition where there should be). I would suggest the authors enlist a native English speaker who is in the field of genetics to carefully overhaul this entire manuscript to help improve the flow of written English. I will list below several examples of these issues, but this is by no means an exhaustive list.

Response: The manuscript has been subjected to a rigorous process of professional English revision with the objective of enhancing its readability.

Additional integrations by the authors:

  • Please note that we have made some minor integrations in Table 1, column “Facial dysmorphisms”, indicating how data have been reported (photo/reported) for each patient.
  • References in Table 1 have been corrected in the revised version of the manuscript.
  • Reference n 18 has been added.

Round 2

Reviewer 2 Report

Comments and Suggestions for Authors

The manuscript is much improved - only minor concerns remain. There are still a few instances where the English can be smoothed to make reading easier. For example: 

* line 84, informed does not need to be capitalized

* line 111, is this supposed to say MEDLINE? (right now it says EDLINE?)

* line 118, change "previously one" to "had one previous unexplained miscarriage". The timing element (here, "previously") doesn't typically come first in English.

* line 127: change "the" to "an" so it reads: At birth, an echocardiogram

* line 170: This would read much smoother if it said: "Having dismissed the suspicion of a syndrome,..."

* line 178: remove "the" so it reads: ...she could role back to front and grasp...

* line 183: remove "the" so it reads: She lost head control, the ability to roll, and the use of her hands.

* line 198: change the timing element here, too, so it reads: "...from France, who had a previously unexplained miscarriage."

* Line 209: Add "A" so it reads: "A first-trimester ultrasound..."

* Line 240: Add "the" so it reads: "The healthy parents were confirmed to be carriers of the variant."

* Lines 304-307: These make much more sense now!

* Line 392: change "were" to "have been" so it reads: ...behavioural anomalies, which have been less consistently reported."

* Line 430: change "continuum" to "continuous", so it reads: "continuous spectrum"

* Line 444: something is missing here; "multi-systemic" should be followed by a noun. "The association of malformations resulted in multi-systemic WHAT in all cases," Maybe the word "syndromes"? "The association of malformations resulted in multi-systemic syndromes in all cases"

* line 448: this phrase is confusing/redundant and needs fixing: "diaphragm and wall diaphragm abdominal wall". Maybe you mean only: "diaphragm and abdominal wall"?

* Line 451: Change "at" to "to" - To the best of our knowlege...

*line 456: change "for" to "due to" - ...post-natal setting due to developmental concern,...

*line 459: remove the "s" from "epilepsy-gene" - the word panels is already plural, so the "s" is not needed on the word gene

* line 460 - change "the" to "this" so it reads: The comprehensive data collection in this review..."

*line 477-478: this is confusing because the sentence says "unrelated patients" but P3 and P4 ARE related. I think just take the word "unrelated" out here.

* line 478: change "at" to "in a" so it reads: ...P3 and P4 in a homozygous or composite heterozygous state.

* line 483: change "at" to "in a" so it reads: ...been documented in a homozygous state...

There are probably additional places where the English could be streamlined. I'm not sure how much editing the journal itself will do to the text. The authors will want their paper to be as excellent as possible.

Comments on the Quality of English Language

Some work remains to make the English as smooth as possible, in order to make reading the paper easier. 

Author Response

We are grateful to the reviewer for the minor revisions. We have changed the manuscript according to the proposed suggestions.

Review – second round

The manuscript is much improved - only minor concerns remain. There are still a few instances where the English can be smoothed to make reading easier. For example: 

* line 84, informed does not need to be capitalized

Response 1: Thank you for the comment. We have changed according to the suggestion (line 84)

* line 111, is this supposed to say MEDLINE? (right now it says EDLINE?)

Response 2: Thank you for the comment. We have changed according to the suggestion (line 111)

* line 118, change "previously one" to "had one previous unexplained miscarriage". The timing element (here, "previously") doesn't typically come first in English.

Response 3: Thank you for the comment. We have changed according to the suggestion (line 118)

* line 127: change "the" to "an" so it reads: At birth, an echocardiogram

Response 4: Thank you for the comment. We have changed according to the suggestion (line 127)

* line 170: This would read much smoother if it said: "Having dismissed the suspicion of a syndrome,..."

Response 5: Thank you for the comment. We have changed according to the suggestion (line 171)

* line 178: remove "the" so it reads: ...she could role back to front and grasp...

Response 6: Thank you for the comment. We have changed according to the suggestion (line 179)

* line 183: remove "the" so it reads: She lost head control, the ability to roll, and the use of her hands.

Response 7: Thank you for the comment. We have changed according to the suggestion (line 185)

* line 198: change the timing element here, too, so it reads: "...from France, who had a previously unexplained miscarriage."

Response 8: Thank you for the comment. We have changed according to the suggestion (lines 199-200)

* Line 209: Add "A" so it reads: "A first-trimester ultrasound..."

Response 9: Thank you for the comment. We have changed according to the suggestion (line 210)

* Line 240: Add "the" so it reads: "The healthy parents were confirmed to be carriers of the variant."

Response 10: Thank you for the comment. We have changed according to the suggestion (line 241)

* Lines 304-307: These make much more sense now!

Response 11: Thank you for the appreciation.

* Line 392: change "were" to "have been" so it reads: ...behavioural anomalies, which have been less consistently reported."

Response 12: Thank you for the comment. We have changed according to the suggestion (lines 393-394)

* Line 430: change "continuum" to "continuous", so it reads: "continuous spectrum"

Response 13: Thank you for the comment. We have changed according to the suggestion (line 431)

* Line 444: something is missing here; "multi-systemic" should be followed by a noun. "The association of malformations resulted in multi-systemic WHAT in all cases," Maybe the word "syndromes"? "The association of malformations resulted in multi-systemic syndromes in all cases"

Response 14: Thank you for the comment. We have changed according to the suggestion (line 446)

* line 448: this phrase is confusing/redundant and needs fixing: "diaphragm and wall diaphragm abdominal wall". Maybe you mean only: "diaphragm and abdominal wall"?

Response 15: Thank you for the comment. We have changed according to the suggestion (line 449)

* Line 451: Change "at" to "to" - To the best of our knowlege...

Response 16: Thank you for the comment. We have changed according to the suggestion (line 452)

*line 456: change "for" to "due to" - ...post-natal setting due to developmental concern,...

Response 17: Thank you for the comment. We have changed according to the suggestion (line 457)

*line 459: remove the "s" from "epilepsy-gene" - the word panels is already plural, so the "s" is not needed on the word gene

Response 18: Thank you for the comment. We have changed according to the suggestion (line 460)

* line 460 - change "the" to "this" so it reads: The comprehensive data collection in this review..."

Response 19: Thank you for the comment. We have changed according to the suggestion (line 461)

*line 477-478: this is confusing because the sentence says "unrelated patients" but P3 and P4 ARE related. I think just take the word "unrelated" out here.

Response 20: Thank you for the comment. We have changed according to the suggestion (line 478)

* line 478: change "at" to "in a" so it reads: ...P3 and P4 in a homozygous or composite heterozygous state.

Response 21: Thank you for the comment. We have changed according to the suggestion (line)

* line 483: change "at" to "in a" so it reads: ...been documented in a homozygous state...

Response 22: Thank you for the comment. We have changed according to the suggestion (line 479)

There are probably additional places where the English could be streamlined. I'm not sure how much editing the journal itself will do to the text. The authors will want their paper to be as excellent as possible
